# Nsun2 coupling with RoRγt shapes the fate of Th17 cells and promotes colitis

Wen-Lan Yang [1,2,13], Weinan Qiu [1,3,4,13], Ting Zhang[1,5,13], Kai Xu [6,7,8,13], Zi-Juan Gu [3,9,13], Yu Zhou [10], Heng-Ji Xu [1,11], Zhong-Zhou Yang [12], Bin Shen [10], Yong-Liang Zhao [1,11], Qi Zhou [6,7,8], Ying Yang [1,2,7] ✉, Wei Li [6,7,8] ✉, Peng-Yuan Yang [3] ✉ & Yun-Gui Yang [1,2,7] ✉

T helper 17 (Th17) cells are a subset of CD4[+] T helper cells involved in the inflammatory response in autoimmunity. Th17 cells secrete Th17 specific cytokines, such as IL-17A and IL17-F, which are governed by the master transcription factor RoRγt. However, the epigenetic mechanism regulating Th17 cell function is still not fully understood. Here, we reveal that deletion of RNA 5-methylcytosine (m5C) methyltransferase Nsun2 in mouse CD4[+] T cells specifically inhibits Th17 cell differentiation and alleviates Th17 cell-induced colitis pathogenesis. Mechanistically, RoRγt can recruit Nsun2 to chromatin regions of their targets, including *Il17a* and *Il17f*, leading to the transcription-coupled m5C formation and consequently enhanced mRNA stability. Our study demonstrates a m5C mediated cell intrinsic function in Th17 cells and suggests Nsun2 as a potential therapeutic target for autoimmune disease.

The polarization of CD4[+] T helper (Th) cell lineages are orchestrated by specific signaling transduction, local cytokines, and transcription factors[1]. The epigenetic factors, such as histone modifications and DNA methylation, participate in differentiation of distinct Th cell lineages through mediating the expression of specific transcription factors and cytokines[2–5]. Meanwhile, RNA *N*[6]-methyladenosine (m6A), one type of RNA methylation, involves naive T cells proliferation and differentiation[6], regulatory T cells (Treg cell) function[7,8], and T follicular helper (T_{FH}) cell differentiation[9,10] via mediating RNA metabolism. In

addition, m6A regulates antigen-specific CD8[+] T cell antitumor response and T cell activation through modulating the function of dendritic cells[11,12]. Another abundant mRNA modification-m5C also seems to be functionally important in the immune response, as one recent report revealed the association of genes containing hypermethylated m5C in CD4[+] T cells with systemic lupus erythematosus (SLE)[13]. Furthermore, Nsun2 methylates *Il17a* mRNA and promotes its mRNA translation in rat total T lymphocytes upon hyperhomocysteinemia treatment[14]. Although RNA modification has been proved to participate in multiple

[1]Key Laboratory of Genomic and Precision Medicine, Collaborative Innovation Center of Genetics and Development, College of Future Technology, Beijing Institute of Genomics, Chinese Academy of Sciences and China National Center for Bioinformation, Beijing 100101, China. [2]Sino-Danish College, University of Chinese Academy of Sciences, Beijing 101408, China. [3]Key Laboratory of Infection and Immunity of CAS, CAS Center for Excellence in Biomacromolecules, Institute of Biophysics, University of Chinese Academy of Sciences, Chinese Academy of Sciences, Beijing 100101, China. [4]Department of Pulmonary and Critical Care Medicine, Ruijin Hospital, Shanghai Jiao Tong University School of Medicine, Shanghai 200025, China. [5]Key Laboratory of Animal Models and Human Disease Mechanisms of the Chinese Academy of Sciences and Yunnan Province, KIZ-CUHK Joint Laboratory of Bioresources and Molecular Research in Common Diseases, Center for Biosafety Mega-Science, Kunming Institute of Zoology, Chinese Academy of Sciences, Kunming, Yunnan 650223, China. [6]State Key Laboratory of Stem Cell and Reproductive Biology, Institute of Zoology, Chinese Academy of Sciences, Beijing 100101, China. [7]Institute for Stem Cell and Regeneration, Chinese Academy of Sciences, Beijing 100101, China. [8]Beijing Institute for Stem Cell and Regenerative Medicine, Beijing 100101, China. [9]National Laboratory of Biomacromolecules, CAS Center for Excellence in Biomacromolecules, Institute of Biophysics, Chinese Academy of Sciences, University of Chinese Academy of Sciences, Beijing 100101, China. [10]State Key Laboratory of Reproductive Medicine, Nanjing Medical University, Nanjing 211166, China. [11]University of Chinese Academy of Sciences, Beijing 100049, China. [12]State Key Laboratory of Pharmaceutical Biotechnology, Model Animal Research Center, Nanjing University Medical School, 210093 Nanjing, China. [13]These authors contributed equally: Wen-Lan Yang, Weinan Qiu, Ting Zhang, Kai Xu, Zi-Juan Gu. ✉e-mail: yingyang@big.ac.cn; liwei@ioz.ac.cn; pyyang@ibp.ac.cn; ygyang@big.ac.cn

T cell immune processes, whether it directly regulates the maintenance of Th17 cell function has not been studied.

Among the Th cell lineages, Th17 cells, one type of pro-inflammatory CD4+ T cells secreting the cytokines, such as IL-17A and IL-17F, are mainly involved in inflammatory response and numerous types of autoimmune disease[15–18]. Retinoic acid receptor-related orphan receptor gamma t (RoRγt) is the key transcription factor of Th17 cells regulating the Th17 cell differentiation through encoding IL-17 gene[19,20]. Th17 cells illustrate high plasticity being capable of conversion into other distinct Th17-lineage cells with different functions depending on the microenvironment[5,21]. The underlying mechanisms for Th17 cell fate determination remain enigmatic. The potential roles of epitranscriptomic mechanism, such as RNA m5C, for Th17 cells homeostasis deserve to be defined.

In this study, we demonstrate that Nsun2-mediated m5C mod-ification is crucial for Th17 cell homeostasis and promotes colitis pathogenesis in mice. Mechanistically, Nsun2 couples with RoRγt to promote m5C formation on Th17-specific cytokines. Nsun2 deficiency in Th17 cells impairs the stability of m5C-modified mRNAs, such as *Il17a* and *Il17f*, resulting in ameliorated colitis progression induced by Th17 cells.

## Results

### Nsun2 is essential for Th17 cells homeostasis

We first tested the mRNA and protein levels of Nsun2 in different organs from wild-type C57BL/6 mice by RT-qPCR and western blot and found that Nsun2 is highly expressed in immune organs, including lymph node, thymus and spleen (Fig. 1a, Supplementary Fig. 1a). To further determine the function of Nsun2 in immunity, we generated *Nsun2* knockout (*Nsun2−/−*) mouse model using CRISPR-cas9-sgRNAs targeting exon 2 of *Nsun2* locus (Supplementary Fig. 1b) and validated the global depletion of Nsun2 by RT-qPCR and western blot in spleen, lymph node and thymus (Supplementary Fig. 1c, d). Furthermore, we analyzed the main subsets of immune cells potentially regulated by

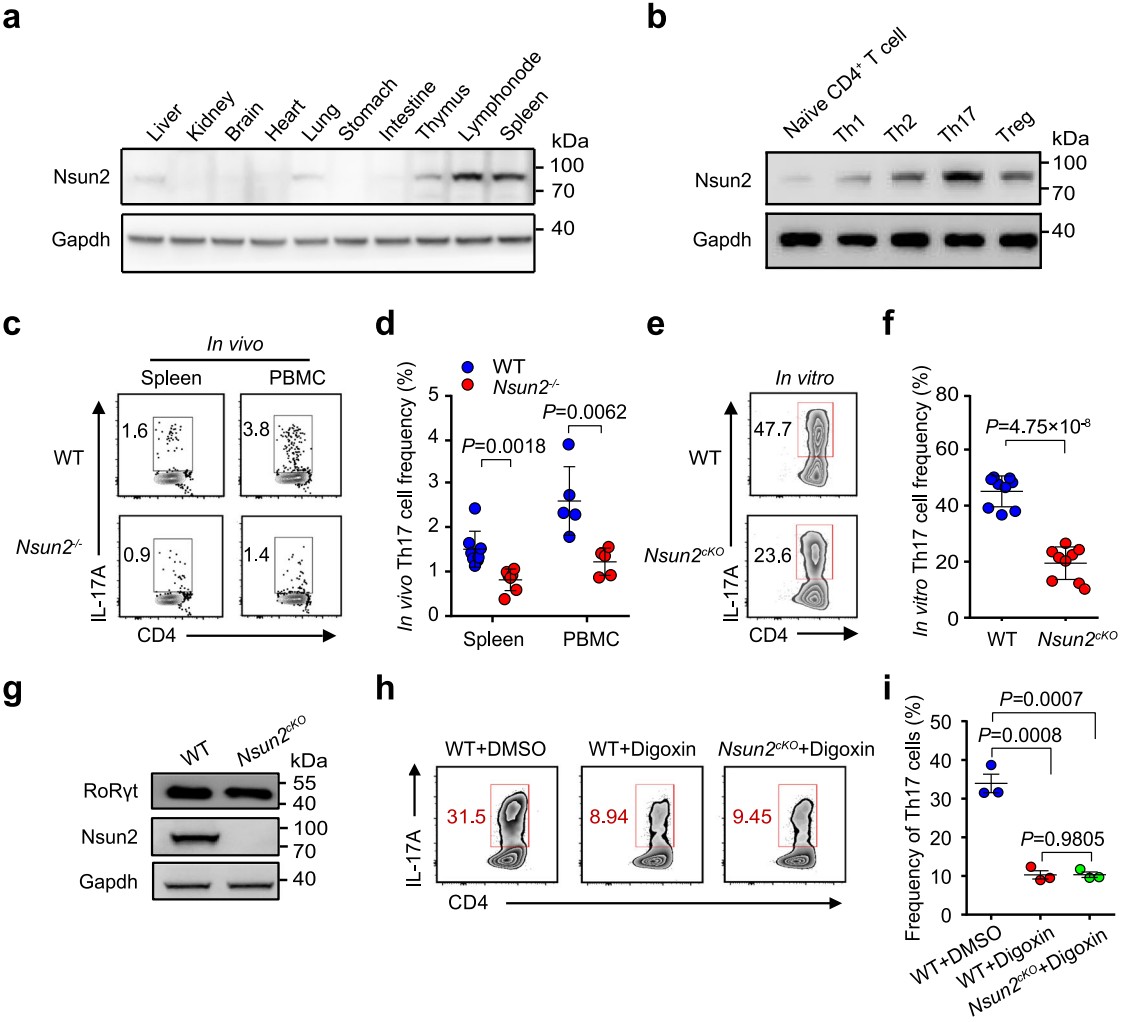

**Fig. 1 | Nsun2 is essential for Th17 homeostasis. a** The protein level of Nsun2 in mouse organs were detected by western blot. **b** Nsun2 protein level in each T cell subsets was investigated by western blot. Sorted naive CD4+ T cells from lymph nodes of wild-type C57BL/6 mice were differentiated under Th1, Th2, Th17, and Treg inducing conditions respectively, and total proteins were then isolated from each T cell subsets. **c**, **d** FACS analyses and statistical (**d**) showing the percentage of CD4+ IL-17A+ Th17 cells in spleen and PBMC from littermate wild-type and *Nsun2−/−* mice. Summary of the frequency of Th17 cells are shown in **d** (*n* = 8 for wild-type group and *n* = 7 for *Nsun2−/−* group in spleen; *n* = 5 for wild-type and *Nsun2−/−* groups in PBMC). **e**, **f** FACS (**e**) and statistical (**f**) analyses showing the in vitro differentia-tion frequency of Th17 cells. Sorted naive CD4+ T cells from wild-type and *Nsun2cKO*

mice respectively were induced under the Th17-inducing conditions. Th17 cells were in vitro differentiated for 4 days. Summary of the frequency of Th17 cells are shown in **f** (*n* = 9 per group). **g** The protein level of RoRγt in Th17 cells was detected by western blot. **h**, **i** FACS analysis showing the in vitro differentiation frequency of Th17 cells under digoxin treatment. Sorted naive CD4+ T cells from wild-type and *Nsun2cKO* mice respectively were treated with DMSO control or digoxin (10 μM) before differentiated under Th17-inducing condition for 4 days. Summary of the frequency of Th17 cells are shown in **i** (*n* = 3 per group). All data are representative of at least two independent experiments. Data were analyzed using two-tailed unpaired Student's *t*-test, Error bars represent mean ± s.e.m (**d**, **f**, **i**). Source data are provided as a Source Data file.

*Nsun2* (Supplementary Fig. 2a), and found that *Nsun2*[-/-] mice maintain normal frequencies of B220[+], CD11b[+], CD4[+] and CD8[+] cells in spleen (Supplementary Fig. 2b, c) and peripheral blood mononuclear cells (PBMCs) (Supplementary Fig. 2d, e), compared to littermate wild-type mice. In addition, the frequencies of activated (CD44[+]CD62L[-]), naive (CD44[-]CD62L[+]) CD4[+] and CD8[+] T cells were not affected by deleting Nsun2 (Supplementary Fig. 2f–i).

Next, we attempted to analyze the effect of Nsun2 depletion on T cell subsets. We first detected a high expression of Nsun2 in Th17 cells both in protein and mRNA levels (Fig. 1b, Supplementary Fig. 3a). Furthermore, we observed that the depletion of Nsun2 specifically dampened the percentage of IL-17A[+] Th17 cells in spleen, PBMCs and lymph node, but not other subsets of T-cells (IFN-γ[+] Th1, IL-4[+] Th2, Foxp3[+] Treg and IFN-γ[+] CD8) (Fig. 1c, d, Supplementary Fig. 3b–h). Given the relatively low frequency of Th17 cells in vivo under naive state, we further validated the Nsun2 regulation on this type of cells using in vitro differentiation. Consistently, only Th17 cells were specifically repressed upon Nsun2 ablation (Supplementary Fig. 4a–c). Moreover, we analyzed the expression changes of the Th17 cell-specific factors upon Nsun2 depletion, and found that the mRNA expression of *Il17a* and *Il17f* decreased significantly (Supplementary Fig. 4d, e).

To define the intrinsic role of Nsun2 on T cell functions, we generated mice with T cell-specific Nsun2 deletion by creating a mouse strain with a floxed *Nsun2* allele (Supplementary Fig. 5a, b). Mice with the floxed *Nsun2* allele were bred with mice bearing a T-cell specific *Cd4-Cre* to generate the *Nsun2*[flox/flox]; *Cd4-Cre* (termed *Nsun2*[cKO] hereafter) mice (Supplementary Fig. 5c). Western blot results indicated the specific depletion of Nsun2 in T cells (Supplementary Fig. 5d). Consistent with the phenotypes observed in the global *Nsun2* knockout mice (Fig. 1c, d), specific ablation of Nsun2 in T cells indeed impeded in vitro Th17 cell differentiation (Fig. 1e, f). Moreover, both the mRNA levels of *Il17a* and *Il17f* (Supplementary Fig. 6a, b), and the secretion of IL-17A and IL-17F (Supplementary Fig. 6c, d) were significantly repressed in *Nsun2*[cKO] Th17 cells.

Since RoRγt is the lineage transcription factor of Th17 cells[19,20], we next compared the expression of RoRγt between in vitro differentiated wild-type and *Nsun2*[cKO] Th17 cells. The results showed that Nsun2 did not affect RoRγt expression in Th17 cells (Fig. 1g and Supplementary Fig. 6e). It has been reported that RoRγt function could be inhibited by digoxin treatment without influencing its expression[22]. We then chose this approach to disable RoRγt and found that Th17 cell in vitro differentiation was significantly suppressed. But RoRγt inhibition on Nsun2-null background didn't cause further decrease in Th17 cell differentiation (Fig. 1h, i). We therefore assumed that Nsun2 and RoRγt coordinate in a same signal axis for controlling Th17 lineage.

## Nsun2 directly interacts with RoRγt

To screen the intrinsically interacting proteins with Nsun2 in Th17 cells, we carried out immunoprecipitation-protein mass spectrum (IP-MS) assay. Intriguingly, we identified the transcriptional factor RoRγt at the top ranking of candidate proteins which strongly interacts with Nsun2 (Fig. 2a, Supplementary Data 1). Furthermore, both the in vivo co-immunoprecipitation (Co-IP) and in vitro GST-pull down assays confirmed the direct interaction between Nsun2 and RoRγt (Fig. 2b, c and Supplementary Fig. 7a). In addition, proximity ligation assay (PLA) and immunofluorescence demonstrated that Nsun2 interacts with RoRγt in the nucleus (Fig. 2d and Supplementary Fig. 7b).

Given the potential role of the transcriptional factor RoRγt, we hypothesized that Nsun2 might regulate the function of RoRγt through affecting its chromatin accessibility in Th17 cells. We thus performed Assay for Transposase-Accessible Chromatin with high throughput sequencing (ATAC-seq) in wild-type and Nsun2 deficient Th17 cells. By averaging signal across all active TSSs, we found that all samples were enriched for nucleosome-free reads at transcription start sites (TSSs) and exhibited characteristic periodicity for nucleosome signals at both

upstream and downstream nucleosomes (Supplementary Fig. 8a). Moreover, the two replicates in both wild-type and *Nsun2*[cKO] mice were highly correlated with each other (Supplementary Fig. 8b, c), and peaks identified from ATAC-seq data in both conditions were preferentially distributed in TSSs (Supplementary Fig. 8d, Supplementary Data 2). In consistent with previous reports[23], the promoters with higher chromatin accessibility were also shown with higher H3K4me3 enrichment and higher gene expression (Supplementary Fig. 8e, f). These findings indicate that ATAC-seq results could efficiently reflect the in-time transcriptional variation upon Nsun2 deficiency.

To test whether RoRγt directly binds to these open chromatin regions which appear to be RoRγt-dependent as identified from our ATAC-seq analysis, we applied the public RoRγt ChIP-seq data (GSE40918)[21] derived from wild-type Th17 cells, and found that RoRγt binding was correlated with the open chromatin regions identified in Th17 cells (Supplementary Fig. 8g). Therefore, we could detect the influence of Nsun2 on RoRγt binding efficiency and the global differential accessibility based on our ATAC-seq when comparing wild-type and *Nsun2*[cKO] Th17 cells. Unexpectedly, we observed almost no significantly dysregulated accessible regions upon Nsun2 knockout (Supplementary Fig. 8h), which was validated by the deoxyribonuclease (DNase) I-treated terminal deoxynucleotidyl transferase–mediated deoxyuridine triphosphate nick end labeling (TUNEL) assay (Supplementary Fig. 8j). Furthermore, no significant change for the footprint of RoRγt across the whole accessible chromatin regions was observed (Fig. 2e). These findings indicate that Nsun2 does not affect the chromatin binding efficiency of RoRγt. To further investigate whether or not Nsun2 depletion affect the transcriptional process, we performed click-it EU labeling and chromatin-associated RNA (caRNA)-seq in wild-type and *Nsun2*[cKO] Th17 cells, and found that Nsun2 depletion didn't affect either the transcriptional initiation (Supplementary Fig. 8k) or the abundance of caRNAs (Supplementary Fig. 8i, l, m, Supplementary Data 3). In summary, although being directly associated with transcription factor RoRγt, Nsun2 depletion has no influence on its chromatin accessibility and global gene transcription.

## Nsun2 couples with RoRγt to catalyze RNA m⁵C formation

Due to the fact that Nsun2 is a m[5]C methyltransferase catalyzing m[5]C formation at RNA levels[24], we speculated that Nsun2 might be recruited to the RoRγt targets to promote m[5]C formation. Nsun2 RIP-seq and m[5]C MeRIP-seq were performed on wild-type Th17 cells, and the results showed that both Nsun2 targeted peaks and the m[5]C modified peaks were mostly distributed on mRNAs, especially in CDS regions with high GC contents (Supplementary Fig. 9a–d, Supplementary Data 3). Moreover, 81.1% of the Nsun2-bound genes were modified by m[5]C modification (Supplementary Fig. 8e) and were also preferentially targeted by RoRγt on their gene body regions (Fig. 2f, Supplementary Data 3). Notably, the promoter binding efficiency of RoRγt was proportional to the enrichment of targeted genes by both Nsun2 and m[5]C (Fig. 2g, h). Next, we found that 56.6% of RoRγt targets were occupied by Nsun2 and modified with m[5]C (Fig. 2i, Supplementary Data 3). Gene ontology (GO) enrichment analysis showed that these genes were significantly enriched for intracellular signaling transduction and T cell receptor signaling pathways (Fig. 2j, Supplementary Data 3), involving the Th17-specific RNAs, such as *Il17a* and *Il17f*.

We next picked the *Il17a* and *Il17f* genes for further verification. In consistent with sequencing results, the chromatin-associated RNAs (caRNAs) of these two genes were bound by both Nsun2 and RoRγt in Th17 cells (Fig. 2k, l). In addition, m[5]C MeRIP-qPCR showed that *Il17a* and *Il17f* RNAs were also modified by m[5]C (Fig. 2m, n). Taken together, these findings suggest that Nsun2 couples with RoRγt to control Th17 cell functional homeostasis through directly methylating Th17-specific transcripts, whereas the chromatin accessibility of RoRγt and the overall transcription initiation are not affected by this process.

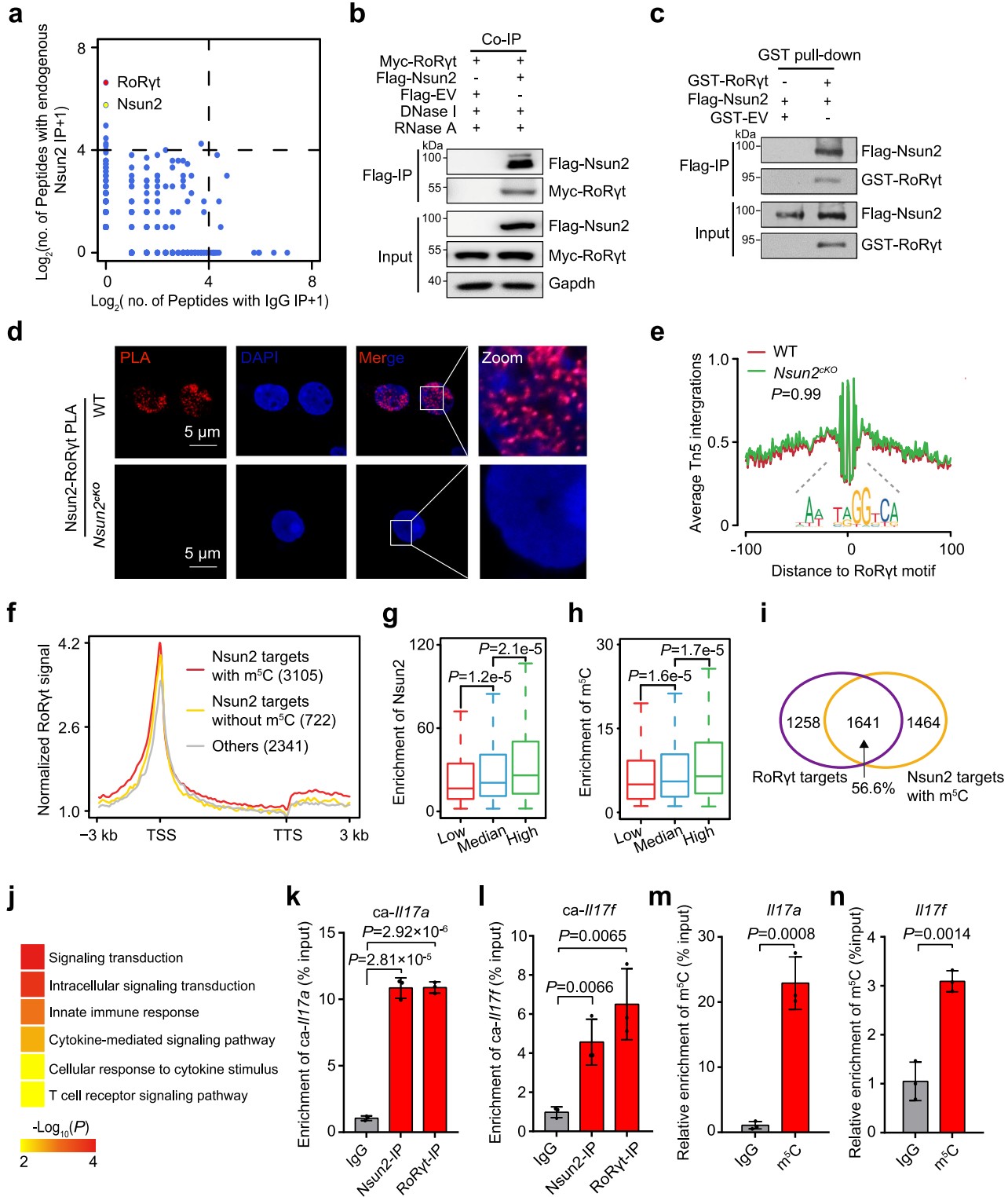

## Nsun2 controls Th17 homeostasis via m⁵C-based mRNA stability

We next determined the regulation of Nsun2 on Th17 cell homeostasis at post-transcriptional level. We first conducted RNA-seq with mRNAs derived from wild-type and Nsun2-deficient Th17 cells, and identified 430 downregulated genes upon Nsun2 deletion, the number of which was much more than that of the upregulated genes (Fig. 3a, Supplementary Data 3). Next, we found that 40% of the downregulated genes were targeted by Nsun2 and modified with m⁵C (Supplementary Fig. 9f). Moreover, 43% of these overlapped genes, including *Il17a* and *Il17f*, were also bound by RoRγt (Fig. 3b and Supplementary Fig. 9g–h,

Supplementary Data 3). In contrast to the unaltered caRNA abundance, we found that the mRNA abundances of *Il17a* and *Il17f* were sharply decreased after Nsun2 knockout (Supplementary Fig. 8l, m), suggesting that m⁵C methyltransferase Nsun2 mainly modulates the mRNA abundance at post-transcriptional level.

To further determine whether or not the intrinsic role of Nsun2 in Th17 cell relies on its m⁵C methyltransferase activity, *Nsun2^{cKO}* T cells were retrovirally infected with wild-type Nsun2 (Nsun2-WT) and enzymatic activity-mutated Nsun2 (Nsun2-C321A) plasmids, respectively, before Th17 induction (Fig. 3c and Supplementary Fig. 10a, b).

**Fig. 2 | Nsun2 couples with RoRγt to promote m5C formation on Th17-related genes. a** IP-MS analysis showing the interacting partners of Nsun2 in induced Th17cells. **b** Co-immunoprecipitation showing the interaction of Nsun2 and RoRγt. **c** GST pull-down assay showing the interaction between Nsun2 and RoRγt. **d** PLA analysis for Nsun2 and RoRγt in induced Th17 cells. Scale bars, 5 μm. DAPI, nuclei. **e** The footprint profiles of RoRγt in Th17 cells with (Red) or without (Green) *Nsun2* depletion. *P* value was determined by two-tailed Student's *t*-test. **f** The metaplots of normalized RoRγt signal along gene bodies from Transcription Start Site (TSS) to Transcription Termination Site (TTS), plus or minus 3 kb. Genes had been divided into three groups including Nsun2 targets with (Red) or without (Gold) m5C and others (Gray). **g, h** Distribution of Nsun2 (**g**) or m5C (**h**) enrichment relative to RoRγt binding efficiency. The Nsun2 targets with m5C were divided into three equal groups according the normalized signal of RoRγt ChIP-seq at their TSSs (±3 kb), each group with 1035 genes. Using the associated IP and input samples, the overall Nsun2 or m5C enrichment distributed in each gene was respectively evaluated by MACS2. *P* value was determined by two-way Mann-Whitney U test and only value between Q1-1.5*IQR and Q3 + 1.5*IQR has been shown in boxplot. **i** Venn diagram showing the overlap between RoRγt targets and Nsun2 targets with m5C. **j** Heatmap displaying gene ontology biological processes enriched in the genes with m5C which were also bound by Nsun2 and RoRγt. *P* value was determined by right-sided Fisher test. **k, l** RT-qPCR showing the binding efficiency of Nsun2 and RoRγt on chromatin-associated *Il17a* (**k**) and *Il17f* (**l**) RNAs in induced Th17cells (*n* = 3 per group). **m, n**, m5C MeRIP-qPCR analysis showing the m5C enrichment in *Il17a* (**m**) and *Il17f* (**n**) mRNAs (*n* = 3 per group). IgG served as negative control. All data are representative of at least two independent experiments. Data are analyzed using two-tailed unpaired Student's *t*-test. Error bars represent mean ± s.e.m. **k–n** Source data are provided as a Source Data file.

Interestingly, we found that overexpression of Nsun2, but not its mutant Nsun2-C321A, can rescue the decreased mRNA levels of *Il17a* and *Il17f* in Th17 cells (Fig. 3d and Supplementary Fig. 11a). Consistently, the decrease of IL-17A⁺ Th17 cells in *Nsun2^cKO* was also rescued by wild-type Nsun2, but not its mutant (Fig. 3e, f). These findings suggest that the intrinsic methyltransferase activity of Nsun2 is essential for Th17 cell generation and functional maintenance.

Recently, Nsun2-mediated m5C modification has been shown to regulate various biological progresses including early embryo maternal-to-zygotic transition and tumorigenesis, through modulating mRNA stability[25,26]. To understand whether Nsun2-mediated m5C influences the mRNA stability of Th17 lineage markers, we tested the half-life of *Il17a* and *Il17f* mRNAs in wild-type and *Nsun2^cKO* Th17 cells under the condition of inhibited transcription by actinomycin D treatment. As expected, loss of Nsun2 significantly accelerated the degradation of both *Il17a* and *Il17f* mRNAs (Fig. 3g and Supplementary Fig. 11b). To further explore the role of m5C modification in maintaining the mRNA stability, we employed m5C-MeRIP RT-qPCR to validate the m5C modification site on C424 of *Il17a* mRNAs that is reported to be a homology site with C367 of *Il17a* mRNAs in rat[14] (Supplementary Fig. 11c). Two minigene reporters: *Il17a*-C (with unmodified C) and *Il17a*-m5C, were synthesized according to the reported m5C-modified cytosine at position 424 on *Il17a* mRNA (Fig. 3h). Th17 cells were transfected with the same amount of each reporter RNA, and then collected at different time points (0, 20, and 40 min) for RT-qPCR analysis. The results showed that the *Il17a*-m5C reporter RNA was more stable than the *Il17a*-C reporter RNA in both wild-type (Fig. 3i), and *Nsun2^cKO* Th17 cells (Supplementary Fig. 11d). Taken together, our findings reveal that Nsun2 maintains Th17 cell homeostasis through regulating the stability of key cytokine mRNAs dependent of m5C modification.

**Targeting Nsun2 in T cells ameliorates colitis development**
Th17 cells have been reported to infiltrate massively into the inflamed intestine and secrete pro-inflammatory cytokines of IL-17A for recruiting neotrophils, which triggers and amplifies the inflammatory pathogenesis in inflammatory bowel disease (IBD) patients[27,28]. To dissect the physiological and pathological functions of Nsun2 in Th17 cells, we induced colitis disease with DSS in *Nsun2^−/−* and wild-type littermates. Notably, compared to the wild-type mice, *Nsun2^−/−* mice had a restricted development of colitis with ameliorated phenotypes of weight decrease (Supplementary Fig. 12a), colon shortened (Supplementary Fig. 12b, c), colon inflammatory infiltration and tissue damage (Supplementary Fig. 12d, e). Moreover, we observed that Nsun2-specific deletion in T cells also relieved occurrence of DSS-induced colitis (Fig. 4a–c and Supplementary Fig. 12f, g). Next, we analyzed the percentage of T cell subsets in mesenteric lymph node (mLN) and spleen. Consistent with the phenotypes observed in mice under normal conditions (Fig. 1c–f), the frequency of Th17 cells was significantly decreased in *Nsun2^cKO* mice when compared to control

mice (Fig. 4d), while minor or unaltered frequencies were shown in Th1, Th2 and Treg cells (Fig. 4e–g and Supplementary Fig. 13). Furthermore, we observed a decreased Th17 cell infiltration in colons of *Nsun2^cKO* mice compared to wild-type mice (Fig. 4h, i), indicating that targeting Nsun2 in T cells mitigates the progress of colitis through dampening Th17 cell function.

To further confirm the repressive role of Nsun2-deletion in T cell-mediated colitis development, we transferred wild-type or *Nsun2^cKO* CD4⁺CD25⁻CD45RB^hi naive T cells into *Rag1^−/−* mice to induce CD45RB^hi adoptive transfer mouse colitis, and found that mice receiving *Nsun2^cKO* naive T cells didn't show any disease phenotype up to 12 weeks after transfer. In contrast, mice receiving wild-type naive T cells began losing weight in the sixth week after transfer and exhibited typical features of colitis, with shortened colon, severe colonic inflammation and disrupted colon structure (Fig. 4j, Supplementary Fig. 14a–f). Furthermore, we also discovered *Nsun2^cKO* naive T cells significantly reduced Th17 cell infiltration in lamina propria of intestinal in the recipient mice compared to wild-type in CD45RB^hi adoptive transfer mouse colitis, whereas no or slight changes were observed in the frequencies of Th1, Th2, and Treg cells (Fig. 4k, Supplementary Fig. 14g). Consistently, blockage of either IL-17A alone or both IL-17A and IL-17F by the specific antibodies in mice receiving wild-type or *Nsun2^cKO* naive CD45RB^hi T cells protected the recipient mice from colitis development with normal phenotypes in body weight, colon macrograph and length, and colon tissue structure (Supplementary Fig. 14a, d and e, Fig. 4j, Supplementary Fig. 15a–c). Collectively, these data support that targeting Nsun2 in T cells ameliorates colitis development.

**Nsun2 targets IL-17A/IL-17F during colitis progression**
We further investigated the function of IL-17A/IL-17F in DSS-induced colitis mouse model. The results showed that neutralization of IL-17A, IL-17F, or both substantially alleviated the DSS-induced colitis development in wild-type mice, with more obvious effect when simultaneously blocking both (Supplementary Fig. 16a–f). In contrast, the alleviation of DSS-induced colitis progression in *Nsun2^cKO* mice could be reversed by injecting recombinant mouse-IL-17A (rIL-17A), IL-17F (rIL-17F), or both (Fig. 5a–f). Collectively, these findings demonstrate that both IL-17A and IL-17F serve as the downstream targets of Nsun2 in Th17 cells and participate in the progression of colitis.

**Nsun2 null reverses the pathological changes of IL-17R⁺ cells**
In order to uncover the regulatory pathway of Nsun2 in colitis microenvironment, we applied single-cell RNA-seq analysis with colons from wild-type and *Nsun2^cKO* mice. A total of 28,495 cells were identified for subsequent analyses (Supplementary Data 4). To define each cell type, we first processed the sequencing data using the DoubletFinder and Seurat R packages for quality control, normalization, batch effect correction, and clustering (see Methods), and then annotated each cell

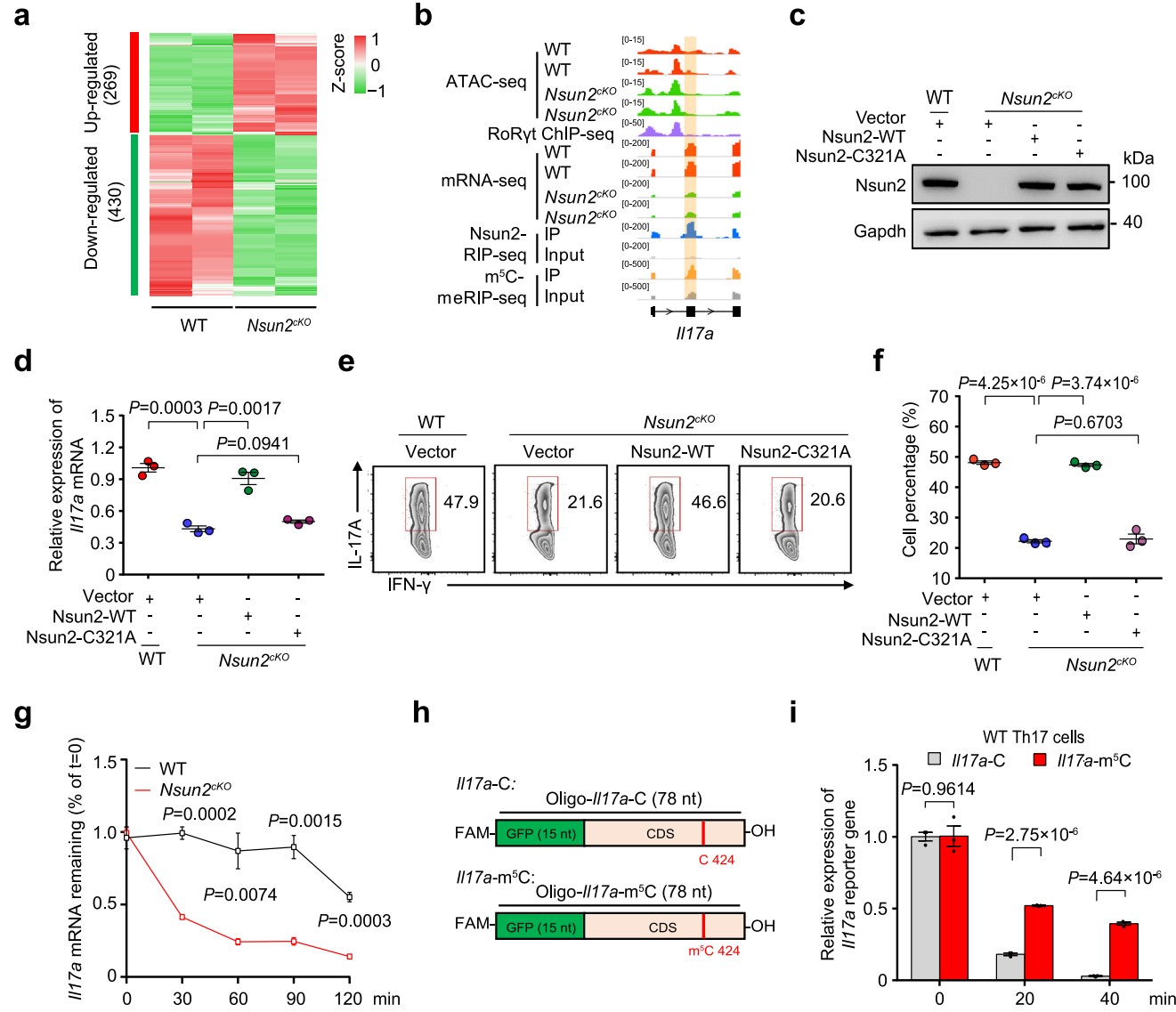

**Fig. 3 | Nsun2 controls Th17 cell generation via m5C-mediated mRNA stability.**
**a** Heatmap showing differentially expressed genes in Th17 cells between wild-type and *Nsun2^cKO* mice. **b** IGV tracks displaying ATAC-seq, RoRγt ChIP-seq (GSE40918), mRNA-seq, Nsun2 RIP-seq (IP and input) and m5C MeRIP-seq (IP and input) enrichment near *Il17a*. **c–f** Verification experiment of rescuing wild-type (Nsun2-WT) and methyltransferase activity deficient mutant (Nsun2-C321A) of Nsun2 in Th17 cells. Sorted naive CD4⁺ T cells from wild-type and *Nsun2^cKO* mice respectively were activated for 1 day, then infected with retrovirus (RV)-mediated Nsun2-WT or Nsun2-C321A expression following induced under Th17-inducing condition for another 3 days. Vector as control. Cell lysates were carried out western blot (**c**). mRNA level of *Il17a* was detected by RT-qPCR (*n* = 3 per group) (**d**), and Th17 differentiated frequency was analyzed by FACS (**e**), and summary of the frequency

of Th17 cells (*n* = 3 per group) (**f**). **g** The half-lives of *Il17a* in Th17 cells of wild-type and *Nsun2^cKO* were detected by RT-qPCR. The residual RNAs were normalized to the values at 0 min (*n* = 3 per group). **h** Schematic representation of the synthesized reporter RNA based on *Il17a* mRNA. *Il17a*-C and *Il17a*-m5C contained 15 nt of GFP sequence and 63 nt of *Il17a* RNA with or without a m5C modification site at 424 of cytosine on *Il17a* mRNA, and the definite sequence information was shown in Supplementary Data 7. **i** The RNA abundance of *Il17a*-m5C (Red) and *Il17a*-C (Gray) in Th17 cells were detected by RT-qPCR, and normalized to the value at 0 min. *Gapdh* serves as an internal control (*n* = 3 per group). All data are representative of at least two independent experiments. Data are analyzed using two-tailed unpaired Student's *t*-test. Error bars represent mean ± s.e.m (**d**, **f**, **g** and **i**). Source data are provided as a Source Data file.

type based on the known markers. In total, the single-cell profiles partitioned into 15 subsets with the cell counts from 66 to 5253 (Fig. 6a, b, Supplementary Data 4). The 15 subsets include 2 epithelial cells (enterocytes and goblet cells), 3 stromal cells (endothelial cells, fibroblasts, and stromal cells), 4 B cells (pro/pre B cells, plasmablasts, and plasma cells), 2 myeloid cells (neutrophils and monocytes), 3 T cells (CD4⁺ T cells, CD8⁺ T cells and γδT cells) and 2 innate lymphoid cells (ILC1 and ILC2) subsets (Supplementary Fig. 17a). Among them, epithelial (24.4%) and B cells (25.8%) were two most enriched subsets. To delineate the dynamics of cell type composition in response to colitis and Nsun2 depletion, we compared the proportion of each cell

type among the WT + H₂O, WT + DSS, and *Nsun2^cKO* + DSS groups (Supplementary Data 4). In wild-type group, we found that 9 of the 15 cell types were aberrantly changed (Supplementary Fig. 17b) during colitis, including fibroblasts, enterocytes, goblet cells, neutrophils, monocytes, CD4⁺ T cells, γδ T cells, ILC1 and ILC2. Moreover, these abnormal changes could be mostly reversed by Nsun2 depletion (Fig. 6c and Supplementary Fig. 17b). For example, the proportions of fibroblasts, enterocytes and goblet cells were decreased during colitis and increased upon Nsun2 depletion, indicating that Nsun2-deletion may prevent stromal and epithelial cells from exhaustion during colitis development. In contrast, the increases of neutrophils and monocytes

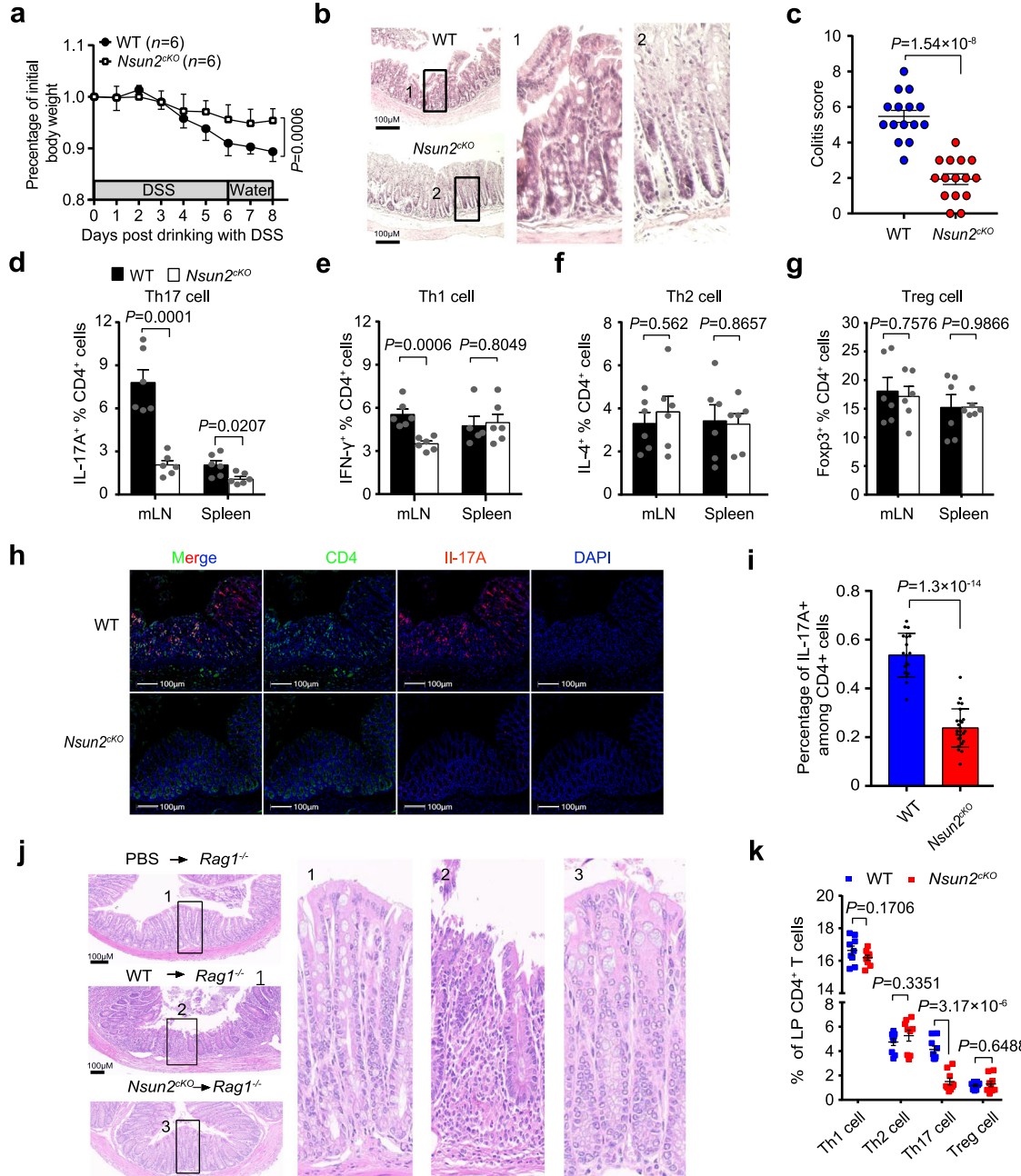

**Fig. 4 | Targeting Nsun2 in T cells ameliorates Th17-mediated autoimmunity.** **a**–**i** The phenotype of wild-type and *Nsun2^cKO^* mice under DSS-induced. Littermate wild-type (*n* = 6) and *Nsun2^cKO^* (*n* = 6) mice were treated with 3% DSS. Body weight changes (**a**), hematoxylin-eosin (H&E) staining indicated the severity of colons damage and inflammatory infiltration (**b**), and colitis score (*n* = 15 image per group) (**c**). FACS showing the T cell subsets frequencies of mesenteric lymph nodes and spleen, including Th17 cell (*n* = 6 per group) (**d**), Th1 cell (*n* = 6 per group in mLN; *n* = 5 for WT and *n* = 6 for *Nsun2^cKO^* in spleen) (**e**), Th2 cell (*n* = 6 per group) (**f**), and Treg cell (*n* = 6 per group) (**g**). **h, i** Immunofluorescence assay showing Th17 infiltration in the colons after DSS-induced colitis. Scale bar, 100 μm. Statistical analysis of the frequency of IL-17A⁺ CD4⁺ T cells in colonic CD4⁺ T cells from immunofluorescence image (*n* = 18 for WT; *n* = 25 for *Nsun2^cKO^*) (**i**). **j, k** The phenotype of wild-type and *Nsun2^cKO^* naive T cell adoptively transferring into *Rag1^-/-^* recipient mice. Representative hematoxylin-eosin staining indicated the severity of colons damage and inflammatory infiltration (**j**). Flow cytometry analysis of IFNγ⁺CD4⁺ T, IL-4⁺CD4⁺ T, IL-17A⁺CD4⁺ T, and Foxp3⁺CD4⁺ T cells frequency in each group recipient wild-type or *Nsun2^cKO^* mice (*n* = 9 per group) (**k**). All data are representative of at least three independent experiments. *P* values were analyzed by two-tailed unpaired Student's *t*-test (**a**, **c**–**g**, **i** and **k**). Error bars represent mean ± s.e.m. Source data are provided as a Source Data file.

in the colitis were also inhibited by Nsun2 depletion, suggesting that Nsun2 may also participates in inflammatory regulation. Intriguingly, we found that all the rescued cell types express IL-17 receptors (Fig. 6d). These results demonstrated that T cell specific depletion of Nsun2 impedes colitis development by preferentially reversing the pathological changes of cells expressing IL-17 receptors, including fibroblast, enterocytes, goblet cells, neutrophils and monocytes.

## Discussion

Nsun2, as one main m⁵C methyltransferase, is widely expressed in various tissues and organs during early embryogenesis[29], and functions in germ cell development[30], epidermal stem cell self-renew[31], neuroepithelial stem cell differentiation[32] and tumor progression[25,33,34]. Here, we indicate that Nsun2 is highly expressed in Th17 cells, and deletion of Nsun2 significantly dampens the frequency of Th17 cells in

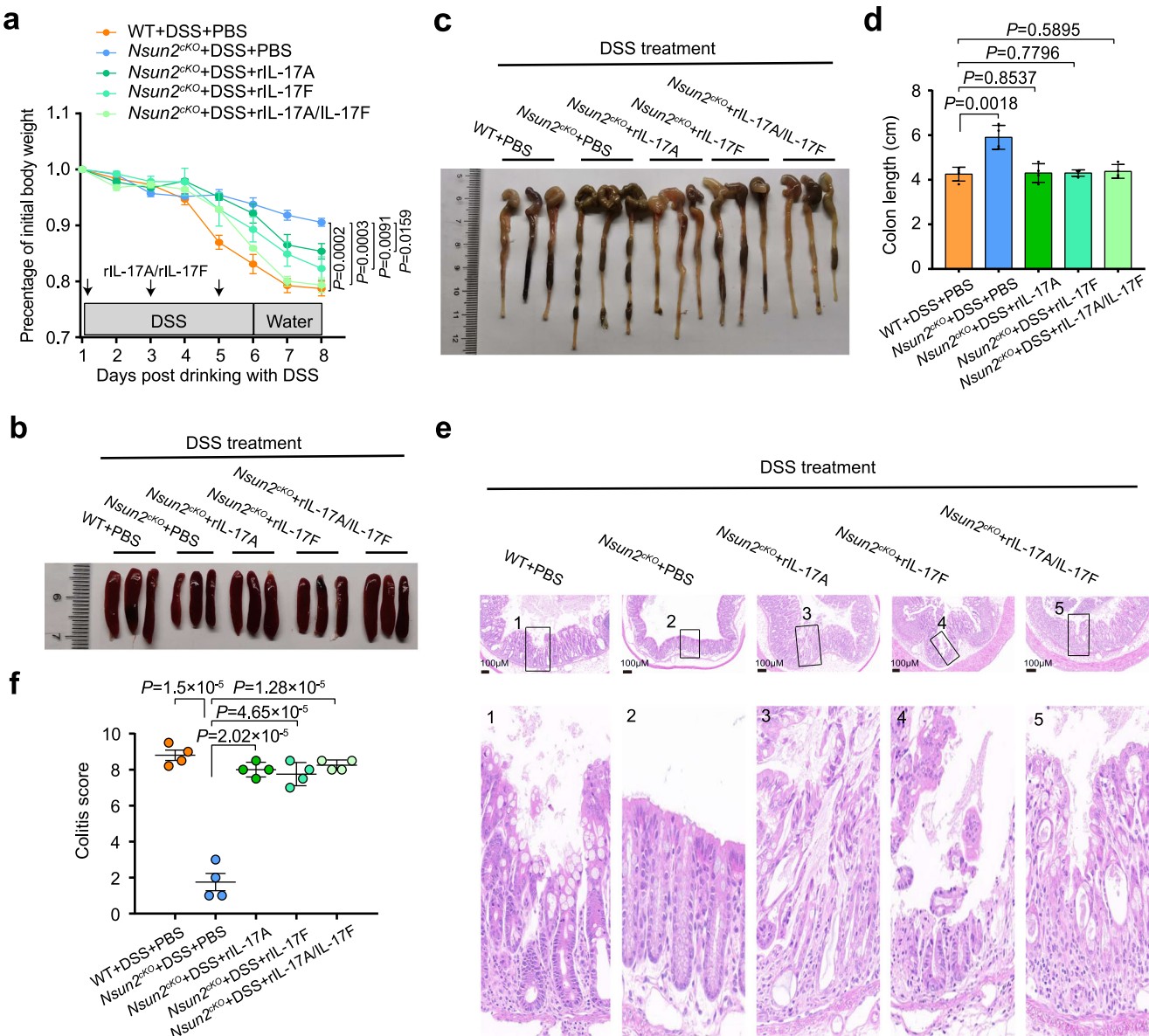

**Fig. 5 | Replenishment of rIL-17A/rIL-17F in DSS-induced Nsun2cKO mice reversed colitis development. a–f** The wild-type and *Nsun2^cKO* mice were treated with 3% DSS in drinking water. *Nsun2^cKO* mice injected with recombinant mouse-IL-17A (rIL-17A) (500 ng/mouse) or IL-17F (rIL-17F) (500 ng/mouse) alone, or both rIL-17A (500 ng/mouse) and rIL-17F (500 ng/mouse) once every other day. Four mice were used in each group. Body weight changes (**a**), representative macrograph of mice, spleens (**b**) and colons (**c**), the colon length (n = 4 per group) (**d**), hematoxylin-eosin staining indicated the severity of colons damage and inflammatory infiltration (Scale bar, 100 μm) (**e**), and colitis score (n = 4 per group) (**f**). *P* values are calculated by using two-tailed unpaired Student's *t*-test (**a**, **d**, **f**). All data are representative of two independent experiments. Error bars represent mean ± s.e.m. Source data are provided as a Source Data file.

mice. Mechanistically, RoRγt directly recruits Nsun2 to mediate co-transcriptional m⁵C formation on Th17-specific cytokine mRNAs including *Il17a* and *Il17f*, which affects both the mRNA stability and the subsequent secretion of IL-17A and IL-17F. Furthermore, Nsun2 depletion impairs the cell-cell communication between Th17 cells and IL-17 receptor-expressing cell clusters, further restricting the progression of colitis.

m⁵C is a crucial post-transcriptional modification that has been reported to regulate many aspects of RNA metabolism, such as mRNA export, stability, and translation, thereby mediating numerous biological processes[24–26,35]. One recent study has shown that Nsun2 expression and m⁵C levels were downregulated in CD4⁺ T cells from patients with systemic lupus erythematosus (SLE), however, upregulated genes with hypermethylated m⁵C were shown to disrupt immune system and promote the flares and remission of SLE patients[13]. Moreover, Nsun2-mediated m⁵C regulates hyperhomocysteinemia-induced

upregulation of IL-17A by promoting its mRNA translation in rat total T lymphocytes[14]. In our study, we reveal that m⁵C specifically involves in governing Th17 cell homeostasis through maintaining Th17 cell-associated mRNAs stability.

Th17 cell is one lineage of T helper cells originated from naive CD4⁺ T cells, and secretes pro-inflammation cytokines, such as IL-17A and IL-17F, participating in inflammatory response, autoimmunity, and transplant rejection[17,18]. RORγt is the key transcription factor to regulate the Th17 cell differentiation through inducing the transcription of the genes encoding IL-17[19,20]. In our study, we found that RORγt recruits Nsun2 to co-transcriptionally catalyze m⁵C formation on CDS regions of *Il17a* and *Il17f* RNAs. In mice, the C424 site of *Il17a* mRNA was homologous with the C367 site of the rat *Il17a* mRNA which has been reported to be methylated by m⁵C[14]. Through reporter gene assay and m⁵C MeRIP qRT-PCR, we verified directly that m⁵C modification on C424 site promotes the stability of *Il17a* RNA. Previous studies

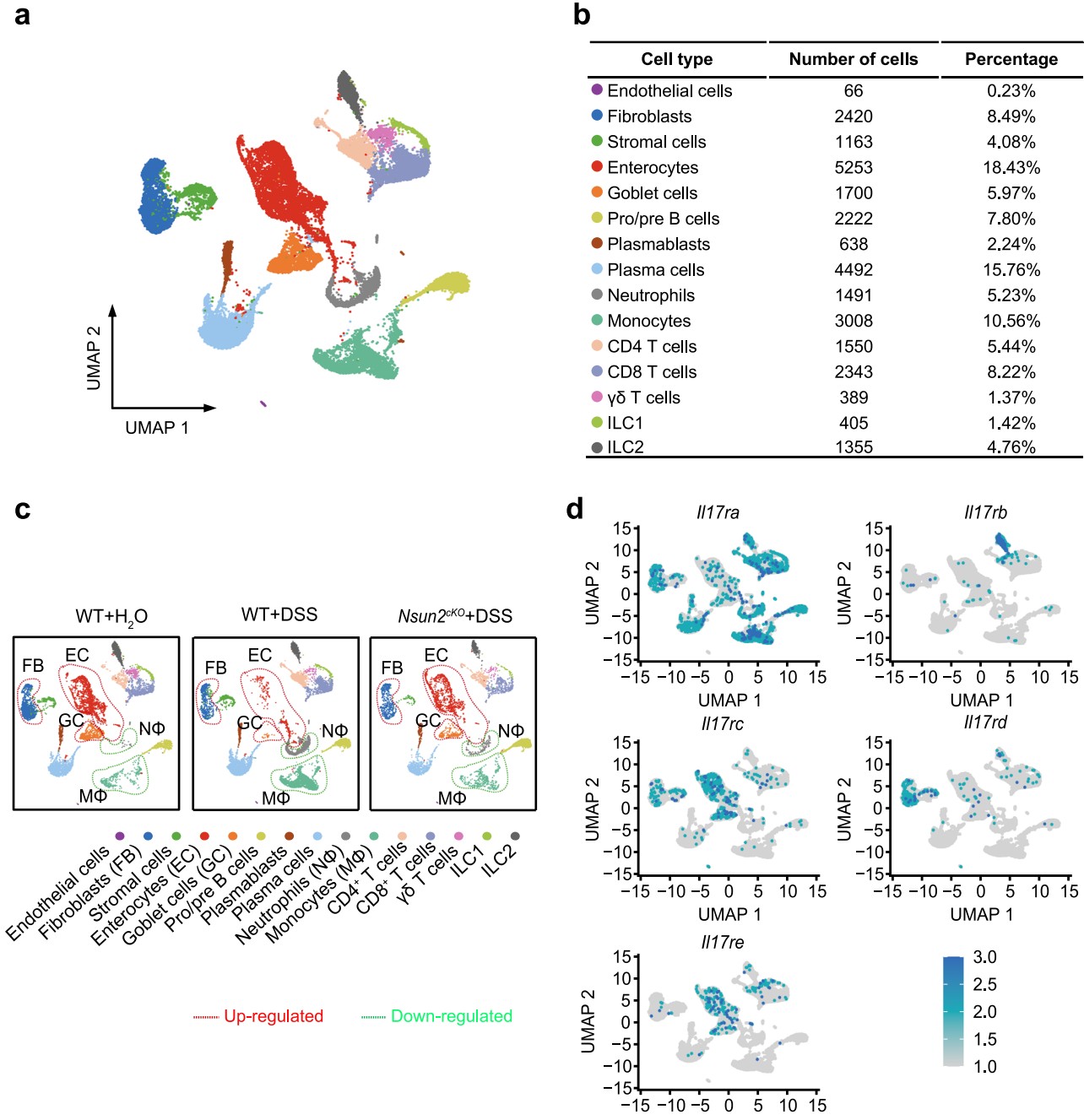

**Fig. 6 | Nsun2 deletion alleviates colitis development by breaking cell-cell communication between Th17 cells and IL-17 receptors-expressing cells.**
**a** UMAP plots of the major clusters. Each point depicts a single cell, colored according to cell types. **b** The chart showing the number and percentage of cell type. **c** UMAP plots showing the different cell types across the three kind of mice colon samples, which respectively under WT + H$_2$O (left), WT + DSS (middle) and *Nsun2^{cKO}* + DSS (right) treatment. The rescued up-regulated cell types were marked in red dash line circles, and the rescued downregulated cell types were marked in green dash line circles. **d** UMAP plots showing expression of IL-17 receptors for rescued cell types. Source data are provided in Supplementary Data 4.

demonstrated that m$^5$C "reader" protein YBX1 recognizes the m$^5$C-modified mRNAs and promotes their stability through recruiting their respective partners, which has been implicated in tumorigenic pathogenesis and embryogenesis[25,26]. As such, m$^5$C-mediated function in Th17 cells might be through a similar mechanism by specific reader(s), which needs further clarification.

Although IL-17A and IL-17F are the important inflammatory cytokines, their function in colitis development is still debatable. Dong et al reported that in DSS-induced acute experimental colitis, IL-17A knock-out increased the colon damage, but IL-17F deficiency resulted in a reduction of colitis development[36]. Also, Iwakura et al. demonstrated

that blockage of IL-17F suppressed the development of colitis in both DSS-induced colitis and T cell-transferring colitis models[37,38]. But other studies have showed that IL-17A, a representative cytokine produced by Th17 cells, has an important role in pathological process of inflammatory diseases[39,40]. Moreover, Pedersen at al showed that simultaneously neutralizing both IL-17A and IL-17F in vivo suppressed development of CD4$^+$CD25$^-$ T-cell transfer colitis, whereas neutralization of IL-17A or IL-17F alone was inefficient[41]. Consistently, Neurath et al. demonstrated that blockage both IL-17A and IL-17F relieves the colitis development, and proposed that combination of anti-IL-17A and anti-IL-17F might be developed as therapeutics for chronic colitis[42]. In Th17-cell related

inflammatory diseases, including psoriasis and psoriatic arthritis (PsA), dual blocking of IL-17A and IL-17 F has been shown to be more effective than IL-17 A blockade alone[43]. In support, our findings demonstrated that blocking of both IL-17A and IL-17F had the most prominent effect on alleviation of colitis development, even though interfering either one also showed beneficial outcome.

Inflammatory bowel disease (IBD) occurs in intestinal mucosa resulting from chronic inflammatory with abnormal immunity induced by the integrated effect of multiple factors, such as individual inheritance, environmental change, and intestinal immune disorders, etc[27]. However, the explicit pathogenesis and treatment strategy of IBD still need further exploration. In IBD patients, it has been reported that Th17 cells massively infiltrate the inflamed intestine and secrete pro-inflammatory cytokines of IL-17A and others, triggering and amplifying the inflammatory pathogenesis[27,28]. One recent study uncovered that T cell-specific deficiency of METTL14, one of the methyltransferases of m6A, induces spontaneous colitis in mice through inhibiting RORγt expression in Treg cells and promoting the differentiation of naive T cells into Treg cells[8]. To explore the functions of m5C in colitis, we generated the DSS-induced and CD45RB[hi] adoptive transfer colitis models in wild-type and Nsun2-deletion mice. Intriguingly, we found that Nsun2 deletion reduces Th17 cells infiltration in lamina propria of intestinal and mitigates the colitis pathogenesis. Furthermore, scRNA-seq data produced from DSS-treated wild-type and *Nsun2*[cKO] colons revealed that Nsun2 deletion reverses the pathological changes of cells expressing IL-17 receptors, including fibroblast, enterocytes, goblet cells, neutrophils and monocytes. During colitis pathogenesis of wild-type mice, IL-17A released from Th17 cells recruits neutrophils to the locations of active inflammation, and also facilitates other pro-inflammatory reactions through inducing stromal cells to secrete matrix metalloproteinases (MMPs), a type of tissue-degrading enzymes which could cause intestinal enterocyte apoptosis[44,45]. Upon Nsun2 deletion, the expression of inflammatory chemokine IL-17A and IL-17F was decreased, while the physiological status of fibroblasts, intestinal cells, and goblet cells, the pivotal components of the colon, was normal under the DSS induction. Moreover, the critical roles of m5C in mediating Th17 cell homeostasis suggest a potential regulatory function in other autoimmune diseases, such as encephalomyelitis.

In summary, our study demonstrates a vital role of Nsun2 and RORγt, through which the RORγt recruits Nsun2 to the chromatin regions of target genes, leading to the transcription-coupled m5C formation of Th17 cell-specific cytokine mRNAs. The resulted enhancement of mRNA stability and subsequent cytokine secretion facilitate the development of colitis. Moreover, the T cell specific depletion of Nsun2 could decrease the expression of cytokines, which may lead to the blockage of the cell-cell communication between Th17 cells and IL-17 receptors-expressing cells, further restricting the progression of colitis (Fig. 7). This unprecedented RORγt-Nsun2 axis may become a potential valuable therapeutic target for clinical autoimmune disease.

## Methods

### Animals

The construction of *Nsun2* knockout (*Nsun2*[−/−]) mice referenced from the previously report[46,47]. The gRNA target sites (*sgNsun2-1*: 5′-GAA CTCAAGATCGTGCCAGAGGG-3′) and (*sgNsun2-2*: 5′-GTACCCAGTGA TTCTCAGTGTGG-3′) for *Nsun2* were designed using the website (https://zlab.bio/guide-designresources), and the sgRNAs and Cas9 constructed into the T7 vector. Both sgRNAs and Cas9 mRNA were then synthesized by applying the HiScribe T7 Quick High Yield RNA Synthesis Kit HiScribe (Cat.no. E2050S; NEB). Each embryo was microinjected with 100 ng/μl Cas9 mRNA, 25 ng/μl *sgNsun2-1*, and 25 ng/μl *sgNsun2-2*. After microinjection, surviving embryos were implanted into the oviduct of pseudopregnant C57BL/6J female mice. Full-term pups were obtained by natural labor at 19.5 dpc.

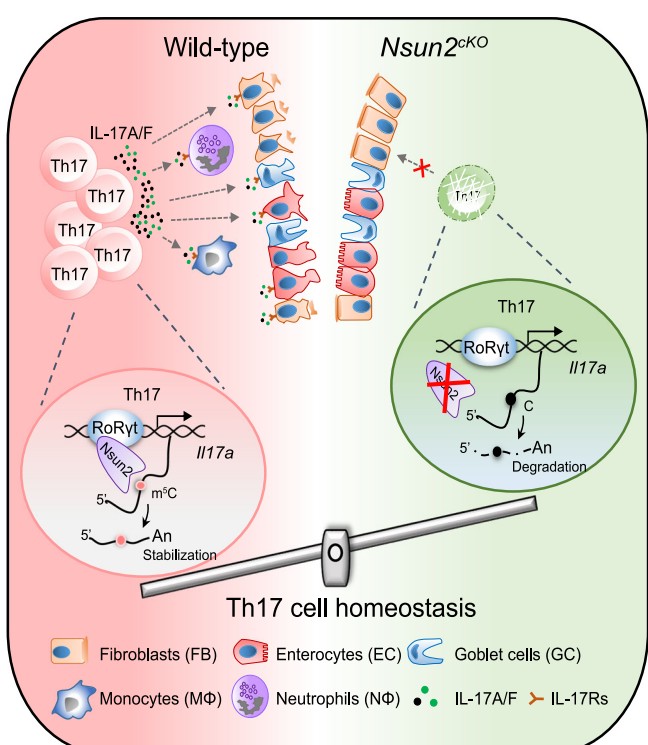

**Fig. 7 | Schematic model of Nsun2 coupling with RORγt regulates Th17 cell homeostasis and promotes the development of colitis.** RORγt, the key transcription factor of Th17 cell and binding to chromatin, directly recruits Nsun2 to in situ catalyze m5C formation on Th17-related cytokine mRNAs including *Il17a* and *Il17f*, which maintains the mRNA stability and pro-inflammatory cytokines subsequent secretion, facilitating occurrence of colitis. However, depletion of Nsun2 restrains the cell-cell communication between Th17 cells and IL-17 receptors-expressing cells, such as, enterocytes, goblet cells, neutrophils and monocytes, etc. to restrict the progression of colitis.

*Nsun2* conditional knockout mice were obtained by crossing *Nsun2*[flox/flox] with *Cd4-cre* mice. For *Nsun2*[flox/flox] mice, the targeted ES cell clone purchased from CAM-SU GRC was used to generate the *Nsun2* KO mice. Then the KO first mice were mated with FLPeR mice to delete the Neo cassette to generate *Nsun2*-floxed mice (Supplementary Fig. 4a, b). *Cd4-cre* mice were purchased from Jackson Laboratory. *Nsun2*[flox/flox]*Cd4-cre* mice were generated by crossing *Nsun2*-floxed mice with *Cd4-cre* mice. All mice were co-housed in the Animal Facilities of the Institute of Biophysics, Chinese Academy of Sciences. Euthanasia by cervical dislocation was performed for all the animals in this study. All investigations involving mice were approved by the Animal Care and Use Committee of the Institute of Biophysics, Chinese Academy of Sciences.

All mice used in this study were kept at C57BL/6J genetic background, and housed under specific pathogen-free (SPF) conditions.

### Flow cytometry and cell sorting

For intracellular cytokines staining, T cells were stimulated in R10 complete medium (RPMI1640 (Cat. no. C11875500BT; GIBCO), 10% FBS (Cat. no. 10091148; GIBCO), 1% Penicilin-Streptomycin (Cat. no. 15140122; GIBCO), 25 mM HEPES (Cat. no. 15630080; Gibco), 1× non-essential amino acid (Cat. no. 11140050; GIBCO) and 0.05 mM β-mercaptoethanol) with phorbol 12-myristate 13-acetate (100 ng/mL) (Cat. no. P8139-1MG; Sigma-Aldrich) and inomycin (500 ng/mL) (Cat. no. 56092-81-0; Sigma-Aldrich) in the presence of brefeldin A (Cat. no. 420601; Biolegend) for 4–6 h. Then the cells were subjected to extracellular staining at 4°C for 30 min, with the Fixation and Permiabilization kit (Cat. no. 88-8824-00; BD Biosciences) according to the instructions of manufacturers. Next, intracellular

staining was carried out with specific antibodies (Supplementary Data 5) at 4 °C for 30 min. Data were acquired by a BD LSRFortessa and analyzed with FlowJo software (FlowJo X 10.0.7r2).

Isolated immune cells from 8 weeks wild-type or *Nsun2*^cKO^ C57BL/6J mice, and then performed extracellular staining at 4 °C for 30 min. Naive T cells (CD8⁻CD4⁺CD25⁻CD62LʰⁱCD44ˡᵒ) were sorted by BD Aria III cell sorter.

## Cell culture

HEK293T (Cat. no. CRL-3216; ATCC) and Plat.E cell lines[48] (provided by Pengyuan Yang's lab) were maintained in high-glucose DMEM (Cat. no. C11995500BT; Gibco) supplemented with 10% FBS. The cells were incubated at 37 °C in a humidified chamber containing 5% $CO_2$. Cell lines used in this study were authenticated with viability, morphology, karyotyping, and STR profiling by the supplier.

## Murine T cells in vitro differentiation

The cell culture and induced differentiation of T helper cell subsets were performed according to the previous report[48]. In briefly, lymphocytes were isolated from the lymph nodes or spleen of mice and naive T cells (CD4⁺CD8⁻CD62LʰⁱCD44ˡᵒCD25⁻) were sorted by BD Aria III cell sorter. T cells were activated with anti-CD3 (BE0002; 2 µg/mL; Bioxcell), anti-CD28 (BE0015-1; 1 µg/mL; Bioxcell). For Th1 cell differentiation, cultures were supplemented with IL-12 (419-ML; 10 ng/mL; R&D system) and anti-IL-4 (504101; 10 µg/mL; Biolegend). For Th2 cell differentiation, cultures were supplemented with IL-4 (404-ML, 20 ng/mL; R&D system) and anti-IFN-γ (505843; 10 µg/mL; Biolegend). For Th17 cell differentiation, cultures were supplemented with IL-23 (1887-ML-010/CF; 20 ng/mL; R&D system), TGF-β1(7754-BH; 1 ng/mL; R&D system), IL-6 (216-16, 60 ng/mL; PeproTech), 10 µg/mL anti-IL-4 and 10 µg/mL anti-IFN-γ. For Treg cell differentiation, cultures were supplemented with 1 ng/mL TGF-β1 and IL-2 (212-12; 0.4 ng/mL; PeproTech). Treg, Th1, Th2 and Treg cells were stimulated for 3 days and Th17 cells were cultured for 4–5 days in vitro. All these T cells were cultured in R10 complete medium.

## Transfection

10 µg DNA for each plasmid was used to transfect HEK293T or Plat-E cells with Lipofectamine 2000 (Cat. no. 11668027; Thermo Fisher) for 48 h in 10-cm dishes, according to the manufacturer's instruction. Cell lysates were isolated and analyzed by immunoprecipitation and western blot.

## Retroviral transduction

Wide-type mouse *Nsun2* and *Nsun2*-C321A were respectively subcloned into pMX-IRES-GFP vector to generate pMX-*Nsun2*-IRES-GFP and pMX-*Nsun2*-C321A-IRES-GFP plasmids. Plat.E packaging cells were transfected with Lipofectamine 2000. The cells were cultured for 48–72 h. The supernatant containing retrovirus was then collected and concentrated by ultracentrifugation of $100,000 \times g$ for 2 h at 4 °C. Activated naive T cells were transduced with retroviral supernatant via low-speed spin ($400 \times g$) for 60 min at 30 °C. The cells were then stimulated under T cell subsets-inducing conditions. The T cells were harvested on day 5 for intracellular staining, FACS analysis, ELISA, and RT-qPCR.

## Enzyme-linked immunosorbent assay (ELISA)

The supernatants were collected after T cell subsets in vitro differentiation, and then the supernatants or serums were analyzed with mouse IL-17A ELISA Kit (Cat. no. DKW12-2170-096; DAKEWE) and mouse IL-17F ELISA Kit (Cat. no. GWB-SKR182; GenWay) according to the manufacturer's instructions.

## Immunoprecipitation and mass spectrometry (IP-MS)

The sorted wild-type naive CD4⁺ T cells were differentiated under Th17-inducing conditions for 4 days. Th17 cells were harvested and pelleted by centrifuging 5 min at $400 \times g$ and then washed three times with 1×PBS and then lysed in RIPA buffer (150 mM NaCl, 1% NP-40, 0.5% sodium deoxycholate, 0.1% SDS, 50 mM Tris (pH 8.0) and 1 mM phenylmethylsulfonyl fluoride) with Protease Inhibitor Cocktail (Cat. no. P8340-1ML; Sigma-Aldrich) and 1×Halt Phosphatase Inhibitor Cocktail (Cat. no. 78440; Thermo Fisher) on ice for 30 min. The collected cell lysates were incubated with Nsun2 antibody (20854-1-AP; 1:50; Proteintech) overnight, and then added pre-cleared Dynabeads™ protein A/G (Cat. no. 88803; Thermo Fisher) for incubation for another 3 h. Finally, washed the beads four times with RIPA buffer. Samples were subjected to NuPAGE™ 4–12% Bis-Tris Gel (Cat. No. NP0321BOX; Invitrogen) and visualized by coomassie blue staining. The protein-containing gel was analyzed by using mass spectrometry in the Institute of Biophysics, Chinese Academy of Science. The proteins identified from the Nsun2 IP-MS analysis were listed in Supplementary Data 1.

## Co-Immunoprecipitation (Co-IP)

Wide-type mouse *Nsun2* subcloned into pCDNA3.1-Flag vector to generate pCDNA3.1-Flag-*Nsun2* (Flag-Nsun2), and wide-type mouse *RoRγt* subcloned into pCMV-Myc vector to generate pCMV-Myc-*RoRγt* (Myc-RoRγt) (Supplementary Data 6). HEK293T cells were co-transfected with Myc-RoRγt and Flag-Nsun2 or Flag-EV plasmids for 2 days. The cells were harvested and washed three times with 1×PBS and then lysed in RIPA buffer (150 mM NaCl, 1% NP-40, 0.5% sodium deoxycholate, 0.1% SDS, 50 mM Tris (pH 8.0) and 1 mM phenylmethylsulfonyl fluoride) with Protease Inhibitor Cocktail and 1×Halt Phosphatase Inhibitor Cocktail on ice for 30 min. Lysates were collected by centrifugation at $17,000 \times g$ for 15 min at 4 °C and measured concentration by BCA kit (Cat. no. 23225; Thermo Fisher). Cell lysates were collected and treated with DNase I and RNase A. The collected cell lysates were incubated pre-cleared anti-Flag beads (Cat. no. M8823; Sigma-Aldrich) or anti-Myc-Beads (Cat. no. 88842; Thermo Fisher) for another 3 h. Finally, the beads were washed 4 times with RIPA buffer. The precipitate was collected and detected by western blot. The antibodies were used including anti-Flag (F7425; 1:2000; Sigma-Aldrich), anti-Myc (C3956; 1:2000; Sigma-Aldrich), and anti-Gapdh (5174S; 1:1500; CST).

## In vitro GST pull down assay

GST pull down assay was referred to the previous report[26]. Specifically, the 5 µg purified Flag-Nsun2 (mouse) protein was pre-cleared with glutathione Sepharose beads (Cat. no. 17075601; GE Healthcare) in NETN buffer (100 mM NaCl, 1 mM EDTA, 20 mM Tris-HCl (pH 7.4), 0.5% NP-40) at 4 °C for 1 h. Then pre-cleared Flag-Nsun2 was mixed with GST or GST-RoRγt protein and incubated at 4 °C for 4 h with equal amount of pre-cleared glutathione Sepharose beads. The complex of IPed proteins and beads was washed five times with NETN buffer, then heated with 4×NuPAGE LDS Buffer (Cat. no. NP0007; Invitrogen) at 95 °C for 10 min, and analyzed by western blot. The plasmids information was listed in Supplementary Data 6 and antibodies were listed in Supplementary Data 5.

## ATAC-seq

The procedure of ATAC-seq was modified from the previous report[23] and based on TruePrep DNA Library Prep Kit V2 for Illumina (Cat. no. TD502; Vazyme). Specifically, $2 \times 10^5$ (count as accurately as possible) fresh Th17 cells were washed once with PBS. The cell pellet was resuspended with 6 µL lysis buffer (10 mM Tris-HCl (pH 7.4), 10 mM NaCl, 3 mM MgCl₂, 0.5% NP-40) and then incubated for 10 min on ice. 4 µL 5×TTBL, 5 µL TTE mix V5, and 5 µL NF-H₂O were added to the sample and incubated at 37 °C for 30 min, then terminated by 5 µL 5×TS stop buffer at room temperature for 5 min. The DNA fragment was purified with phenol-chloroform (pH 7.9), precipitated with ethanol, and then dissolved in 29 µL NF-H₂O and transferred to 0.2-mL

PCR tube. The PCR was performed in 5 μL N7XX primer, 5 μL N5XX primer, 10 μL 5×TAB and 1 μL TAE system with 13 cycles using the following conditions: 72 °C for 3 min; 98 °C for 30 s; and thermos cycling at 98 °C for 15 s, 60 °C for 30 s and 72 °C for 3 min; 72 °C 5 min. The libraries were then purified with VAHTS DNA Clean Beads (Cat. no. N411; Vazyme) and sequenced on the Illumina NovaSeq 6000.

## caRNA-seq and caRNA RT-qPCR

caRNA extraction procedures were modified from the previous report[49,50]. Specifically, ~1 × 10[7] Th17 cells were obtained from wild-type and *Nsun2*[cKO] mice, washed twice with 1×PBS. The cell was lysed with 400 μL Igepal lysis buffer (10 mM Tris HCl (pH 7.5), 150 mM NaCl, 0.15% Igepal CA-630, 0.4 μL 20 U/μL SUPERase•In (Cat. no. AM2694; Thermo Fisher), α-Amanitin (Cat.no. 23109-05-9; MCE), and incubated on ice for 5 min. Meanwhile, 1 mL pre-cooled Sucrose buffer (10 mM Tris-HCl (pH 7.5), 150 mM NaCl, 24% Sucrose, 1 μL 20 U/μL SUPER-ase•In) was prepared in a 1.5 mL Protein LoBind EP tube (Cat. no. 0030108302; Eppendorf). The cell lysate was then gently overlaid on top of the Sucrose buffer and centrifuged at 3500 × g for 10 min. The supernatant was removed and cell nuclei pellet was washed twice with 1 mL pre-cooled PBS EDTA (42 mM KH$_2$PO$_4$, 8 mM Na$_2$HPO4, 136 mM NaCl, 2.6 mM KCl, 0.5 mM EDTA, 1 μL 20 U/μL SUPERase•In). The nuclei were then resuspended in 250 μL glycerol buffer (20 mM Tris-HCl (pH 7.5), 75 mM NaCl, 0.5 mM EDTA (pH 8.0), 50% glycerol)), immediately added to 250 μL Nuclear lysis buffer (10 mM Tris-HCl (pH 7.5), 300 mM NaCl, 7.5 mM MgCl$_2$, 0.2 mM EDTA, 1% Igepal CA-630, 1 M Urea, 0.5 μL 20 U/μL SUPERase•In), and mixed by vortexing for 4 s and incubated on ice for 2 min. Afterwards, the lysate was centrifuged at 17,000 × g for 2 min to precipitate the chromatin-RNA complex. The chromatin pellet was briefly rinsed with PBS-EDTA and resuspended in 1 mL Trizol reagent and the following RNA isolation procedures according to the manufacturer's instructions. The remaining DNA was removed by using TURBO™ DNase (Cat.no. AM2239; Invitrogen) at 37 °C for 40 min. The library of caRNA-seq was constructed with KAPA RNA Hyper Kit (Cat. no. KK8544; KAPA) and sequenced on the Illumina NovaSeq 6000. The caRNA RT-qPCR was performed by using ChamQ Universal SYBR qPCR Master Mix (Cat. no. Q711; Vazyme). The primers sequence information was provided in the Supplementary Data 7.

## Nsun2 RIP-seq

Nsun2 RIP-seq procedures were modified from the previous report[26]. Specifically, ~2 × 10[7] Th17 cells were irradiated twice with 0.15 J/cm[2] (Stratagene). The cell was resuspended with 500 μL lysis buffer (50 mM Tris-HCl (pH 7.4), 100 mM NaCl, 1% NP-40, 0.1% SDS, 0.5% Na-Deoxycholate, 1:100 protease inhibitor cocktail and 0.4 U/mL RNase inhibitor), and treated with Micrococcal Nuclease (Cat. no. M0247S; NEB) (1:1×10[6]) at 37 °C for 10 min. Then the sample was centrifuged at 4 °C with 17,000 × g for 20 min to clear the lysate and collect the supernatant. The dynabeads™ protein A was pre-cleared by lysis buffer for three times and resuspended in 100 μL lysis buffer. Eighty microliters of pre-cleared beads was added to Nsun2 antibody (20854-1-AP; 1:50; Proteintech), and the antibody-beads mixture was then incubated at 4 °C for 60 min with gentle rotation. To exclude non-specifically bound proteins, 20 μL of pre-cleared beads was added to the sample lysate at 4 °C for 60 min with gentle rotation. The protein-antibody-beads complex then was resuspended with the pre-cleared sample lysate, and incubated at 4 °C for 4 h with gentle rotation. Then the beads were collected and washed twice with 1 mL high-salt buffer and twice with 1 mL PNK buffer (20 mM Tris-HCl (pH 7.4), 10 mM MgCl$_2$, 0.2% Tween-20). The RNA-protein-beads complex was resuspended with 20 μL dephosphorylation mix (14.5 μL NF-H$_2$O, 4 μL 5×PNK buffer (pH 6.5), 1 μL T4 PNK (Cat. no. M0201; NEB), 0.4 U/mL RNase inhibitor), and incubated at 37 °C with gentle rotation for 20 min. Then the beads were washed once in 1 mL of PNK wash buffer, 1 mL of high-salt buffer (50 mM Tris-HCl (pH 7.4), 1 M NaCl, 1 mM EDTA, 0.1% SDS, 0.5% Na-Deoxycholate, 1% NP-40) and 1 mL of ice-cold 1×PK buffer (100 mM Tris-HCl (pH 7.5), 50 mM NaCl, 10 mM EDTA). The 200 μL PK buffer and 4 μg/μL proteinase K (Cat.no. 3115828001; Roche) were then added to RNA-protein-beads complex and incubated with shaking at 1100 rpm for 40 min at 50 °C. The target RNA was finally isolated by using 200 μL acid phenol:CHCl$_3$ (pH 5.5) and ethanol precipitation. The libraries of Nsun2 RIP-seq were constructed using SMARTer® Stranded Total RNA-Seq Kit v2-Pico Input Mammalian (Cat.no. 634414; Takara) and sequenced on Illumina NovaSeq 6000.

## m5C MeRIP-seq and m5C MeRIP-qPCR

1 μg mRNAs extracted from Th17 cells sorted from wild-type mice were treated with TURBO™ DNase and then fragmented to about 200 nt fragments at 90 °C for 30 s in 10 × RNA fragmentation reagent (Cat. no. AM8740; Ambion). 60 μL Dynabeads™ Protein A was pre-cleared by 500 μL IPP buffer (150 mM NaCl, 0.1% NP-40, 10 mM Tris-HCl (pH 7.4), 0.4 U/mL RNase inhibitor). Three micrograms of anti-m5C (ab10805; 1:50; Abcam) and IgG (A7016; 1:50; Beyotime) were respectively incubated with pre-cleared Dynabeads™ Protein A in 150 μL IPP buffer for 1 h at room temperature. Fragmented mRNAs were then incubated with the prepared antibody-beads mixture at 4 °C for 4 h. The beads pellet was washed three times with IPP buffer and twice with high salt buffer (50 mM Tris, pH 7.4, 1 M NaCl, 1 mM EDTA, 1% NP-40, 0.1% SDS), and then subjected to proteinase K digestion, phenol chloroform extraction and ethanol preparation to recover the bound RNAs. One-tenth of the fragmented RNA was saved as input control. Equal amount of input RNA and IPed RNA was subjected to reverse transcription for the subsequent qPCR. The relative enrichment of m5C in each sample was calculated by normalizing to that of IgG. The primers used for RT-qPCR were listed in Supplementary Data 7.

## RNA half-life assay

Induced Th17 cells were plated on 96-well plates with 0.5 million cells per well. Actinomycin-D (Cat. no. 50-76-0; Sigma-Aldrich)[6] was added to a final concentration of 5 μM, and Th17 cells were collected at different time points ($t = 0, 30, 60, 90, 120$ min) after adding Actinomycin-D. Total RNAs were extracted from Th17 cells and applied for RT-qPCR. For calculation, the data were normalized to the $t = 0$ time point. The primer sequences for RT-qPCR were listed in Supplementary Data 7.

For the half-life assay of m5C- or C-reporter minigenes, the reporter RNAs (Oligo-*Il17a*-m5C or Oligo-*Il17a*-C) form of 52 nt RNA fragment of *Il17a* around C424, and a 15 nt sequence of GFP, which were synthesized by GenScript company (Supplementary Data 7). The equal amount of Oligo-*Il17a*-m5C and Oligo-*Il17a*-C were transfected into 1 × 10[6] Th17 cells by using electroporator (Etta Biotech), respectively. The cells were collected at indicated time points post transfected ($t = 0, 20, 40$ min) for the RNA stability analysis. The RNA stability was detected through measuring the RNA abundance changes over the time by RT-qPCR. The sequences of primers used in this study are listed in Supplementary Data 7.

## Colitis model

The dextran sulfate sodium salt-induced colitis referenced the previous report[40]. Briefly, the 8-week-old C57BL/6 J mice (littermate) were administered 3% DSS (Cat. no. 160110; MP Biomedicals) in their drinking water for 6 days, and changed to distilled water for 2 days. For blockage of IL-17A and IL-17F, the antibodies of IL-17A (BE0173; 250 μg/mouse; BioXcell), IL-17F (BE0303; 250 μg/mouse; BioXcell) or both IL-17A (250 μg/mouse) and IL-17F (250 μg/mouse) were injected into mice by intraperitoneal administration one day before DSS treatment. For replenishment of IL-17A and IL-17F in *Nsun2*[cKO] mice, the recombinant mouse-IL-17A (rIL-17A) (Cat. no. RP02520; 500 ng/mouse; ABclonal), IL-17F (rIL-17F) (Cat. no. RP01147; 500 ng/mouse; ABclonal), or both rIL-17A (500 ng/mouse) and rIL-17F (500 ng/mouse)[42] were injected into mice by intraperitoneal administration once every other day post

DSS treatment. The mice were weighed daily and euthanized for analysis on the last day.

We performed the CD45RB[hi] adoptive transfer colitis as described[51,52]. Briefly, the pure CD4+CD25−CD45RB[hi] naive T cells were sorted from 6-week-old wild-type and *Nsun2*[cKO] C57BL/6 J mice by FACS, washed twice with PBS, and $5 \times 10^5$ cells of each type were intravenously injected into 7-week *Rag1*[-/-] recipient C57BL/6 J mice (GemPharmatech™) through retro-orbital venous sinus, respectively. PBS vehicle injection was used as control. The recipient mice were monitored and weighted each week. For neutralization of IL-17A and IL-17F, the antibodies of IL-17A (50 µg/mouse) and IL-17F (50 µg/mouse)[37] were injected into mice by intraperitoneal administration two times per week.

## Haemotoxylin and eosin staining
Formalin-fixed, paraffin-embedded tissue sections mounted on glass slides were used for H&E staining. The slides were first deparafinzated and then stained with Hematoxylin and Eosin (Cat.no. 60524ES60; YEASEN), respectively. Finally, the tissues were mounted with resins and observed by microscope.

## RNA-seq and RIP-seq data processing
After pre-processing, the remaining reads of all samples were aligned to the mouse genome (mm10) by HISAT2 (version 4.8.5)[53]. To minimize the rate of false positives, only uniquely mapped reads with -q ≥ 20 were kept for the subsequent analysis for each sample. In addition, for mRNA-seq, caRNA-seq, and the inputs of Nsun2 RIP-seq and m[5]C MeRIP-seq, the number of reads mapped to each gene was counted using the HTSeq (version 0.11.2)[54], with parameter '--mode = union and --stranded = no'.

## ATAC-seq data processing
As previously reported[55], all ATAC-seq reads were aligned to mm10 using Bowtie2 (version 7.3.0) with the parameters "-t -q -N 1 -L 25 -X 2000 --no-mixed --no-discordant". ChIP-seq reads were aligned to mm10 with the parameters "-t -q -N 1 -L 25". All unmapped reads, non-uniquely mapped reads, mitochondrial reads, and PCR duplicates were removed. For nucleosome positioning, peak-calling, and footprinting analysis, we have adjusted the read start sites using ATACseqQC[56].

## Nucleosome positioning
To generate the nucleosome position data track, we split ATAC-seq reads into various bins as previously described[57]. Next, reads were analyzed using ATACseqQC with default parameters to get the enrichment signals for nucleosome-free and nucleosome-bound fragments.

## ATAC-seq peak calling and annotation
For ATAC-seq, only reads with high mapping quality score (-q ≥ 20) were included for peak calling and MACS2 (version 2.1.4)[58] were used with parameters "--nomodel -g mm -B -q 10e-5 --keep-dup all". For downstream analysis, the raw data of two replicates, obtained from wild-type or *Nsun2*[cKO] mice, were respectively merged. Only the common peaks identified among two replicates and the associated merged sample were kept and annotated by HOMER program (version 4.7).

## The comparison between ATAC-seq replicates
To accurately evaluate repeatability between different ATAC-seq replicates, the technical differences need to be removed, which generated during sample preparation and sequencing. To solve this problem, we used a trimmed-mean approach[59] with some modifications. Briefly, a set of low-variance peaks ("null" peaks) was identified, and the mean read counts in this set were used to determine the per-sample scale factors. The peaks forming this set were selected by annotating peaks with expression data from wild-type or *Nsun2*[cKO] mice. Peaks containing transcription start sites (±500 bp) for genes that were both expressed (mean RPKM > 5 for each group) and had low variance (variance in RPKM < 0.05) were assumed to accurately reflect the change in read count due to technical differences. These criteria produced a set of 541 null peaks. The sum of reads within these peaks, normalized by the geometric means, was used as scale factors for each sample.

Furthermly, the correlation between biological replicates was calculated as following: the read count of each common peak was calculated using BEDTools (version 2.16.2)[60], normalized to RPKM value using in-house script and finally weighted with scale factor for each sample. Pearson correlation was used for all analysis.

## The correlation between chromatin accessibility and gene expression
In order to accurately estimate the correlation between chromatin accessibility and gene expression in wild-type Th17 cells, only the most abundant transcript for each gene was kept for the downstream analysis. The read counts of ATAC-seq around TSSs (±3 kb) were calculated and then normalized to RPKM values. Only the top 15 K genes with highest RPKM values were used and were then divided into three groups with RPKM values from low to high. Next, H3K4me3 ChIP-seq of Th17 cells were downloaded from GSE40918, the enrichment of H3K4me3 ChIP-seq around TSSs were also calculated as mentioned above. In addition, the average expression for each gene was quantified using mRNA-seq datasets of wild-type Th17 cells.

## TF footprinting
To identify whether the footprints of RORγt is different nor not, the ATAC-seq peaks of wild-type or *Nsun2*[cKO] mice and duplicates-removed bam files were used as input files for Rgt-Hint-ATAC[61] with the parameters "--atac-seq --paired-end --organism=mm10".

## RIP-seq peak calling and annotation
Peaks of Nsun2 RIP-seq and m[5]C MeRIP-seq were called using MACS2 program with default options except for '--nomodel'. The peaks were annotated based on Ensembl (release 68) gene annotation information by applying BEDTools' intersectBed.

Next, we defined a gene set (L1) with relatively high expression across different samples of mRNA-seq and input samples of Nsun2 RIP-seq and m[5]C MeRIP-seq. The criterion of LI was setted as following: (1) mean RPKM > 1 in wild-type replicates of mRNA-seq; (2) RPKM > 1 in all input samples of Nsun2 RIP-seq and m[5]C MeRIP-seq. In this study, only peaks located in L 1 genes were regarded as reliable and were used for subsequent analysis.

## GC content analysis
To evaluate the GC ratio, all peaks identified by Nsun2 RIP-seq and m[5]C MeRIP-seq were respectively used as the target sequences and background sequences were constructed by randomly shuffling peaks on mouse genome using BEDTools' shuffleBed (version 2.16.2). The GC ratio were calculated using R program (version 3.6.0).

## Differential analysis of ATAC-seq and mRNA-seq
To evaluate whether *Nsun2* deficiency change the chromatin accessibility or not, DEseq2[62] was used. Differential peaks were identified using fold-change cutoff = 2, FDR cutoff = 0.001.

Differential expressed genes between wild-type or *Nsun2*[cKO] mice were determined using DEGseq[63], fold-change cutoff =1.2, P value cutoff = 0.05. To remove the noisy caused by low-expression genes, only the differentially expressed genes listed in L 1 were kept for subsequent analysis.

## scRNA-seq data processing and determination of the major cell types
Reads were demultiplexed and mapped to mm10 through Cellranger toolkit (version 2.1.0, 10× Genomics) and subsequent analysis

# Article

were performed with R package Seurat v3[64]. Data filtering was conducted by retaining cells expressed 500 and 4500 genes inclusive, and had mitochondrial content <20%. To account for differences in sequencing depth across cells, UMI counts were normalized by the total number of UMIs per cell and converted to transcripts-per-10,000 before being log transformed (henceforth "Log(TP10K + 1)"). The top 2000 highly variable Genes (HVGs) were selected using the "FindVariableGenes" function with the default parameters. And doubletFinder (version 2.0.2)[65] was employed to exclude doublets. Then all the datasets were integrated using the "FindIntegrationAnchors" and "IntegrateData" function in Seurat. Merged data were scaled to unit variance and zero mean. The dimensionality of data was reduced by principal component analysis (PCA). A K-nearest-neighbor graph was constructed based on the Euclidean distance in PCA space using the "FindNeighbors" function and Louvain algorithm was applied to iteratively group cells together by "FindClusters" function with optimal resolution on the optimal principal components. Visualization was achieved by the UMAP. Finally, specific markers in each cluster were identified by the "FindAllMarkers" function and clusters were assigned to known cell types using the canonic markers.

### Cell type composition variation analysis

The cell type frequencies were obtained by dividing the numbers of cells of each type in the different groups (WT + H$_2$O, WT + DSS, *Nsun2$^{cKO}$* + DSS) by the total number of cells in the same group. Based on these frequencies, the percentage of a given cell type was calculated for each group. The Log$_2$ (fold change) between the WT + H$_2$O and WT + DSS was then calculated to identify the cell types altered during colitis (|Log$_2$ (fold change)| > 0.6), and the Log$_2$ (fold change) between the WT + DSS and *Nsun2$^{cKO}$* + DSS groups was calculated to identify the cell types altered by *Nsun2* depletion (|Log$_2$ (fold change)| > 0.6). In this study, "rescue cell type" was defined as the cell types that were altered during colitis and rescued upon *Nsun2* depletion.

### Statistics and reproducibility

Statistical analysis was performed with Graph Prism 7.0 software. The two-tailed unpaired Student's *t*-test and Mann–Whitney *U* test were used to calculate *P* values. *P* < 0.05 were considered as statistically significant. The correlation between replicates was analyzed using Pearson's test. The survival rate was analyzed by Kaplan–Meier survival curve. All data are representative of at least two independent experiments.

### Reporting summary

Further information on research design is available in the Nature Portfolio Reporting Summary linked to this article.

## Data availability

The ATAC-seq, Nsun2 RIP-seq, m5C MeRIP-seq, caRNA-seq, mRNA-seq, and scRNA-seq data generated in this study have been deposited in Genome Sequence Archive of National Genomics Data Center under accession code CRA005161 (linked to the BioProject with accession No PRJCA006795). The mass spectrometry data of Nsun2-IP have been deposited in the PeptideAtlas with accession number PASS01711. Source data are provided with this paper.

## Code availability

All softwares used in this study are publicly available. The main code has been deposited in github under https://github.com/Candiesbio75/Nsun2-coupling-with-RoR-t-shapes-the-fate-of-Th17-cells-and-promotes-colitis. Source codes are available from the authors upon request.

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

## Acknowledgements

We thank Prof. Huabing Li from Shanghai Jiao Tong University School of Medicine for the scientific discussion. We thank Ji-Feng Wang, Xiang Ding, and Mengmeng Zhang at the laboratory of Proteomics, core facility in the Institute of Biophysics, Chinese Academy of Sciences (CAS) for their technical support of liquid chromatography-mass spectrometry (LC-MS) analysis. We thank Junfeng Hao from Core Facility for Protein Research, Institute of Biophysics, Chinese Academy of Sciences for technical support of intestinal immunofluorescence and analysis. Meanwhile, we also thank Yutian Deng from Core Genomic Facility, Beijing Institute of Genomics, Chinese Academy of Sciences, and Xiao-yan Wang from Key Laboratory of Infection and Immunity in Institute of Biophysics, Chinese Academy of Sciences for the technical support of FACS. This work was supported by the Science Fund for Creative

Research Groups of the National Natural Science Foundation of China (32121001 Y.-G.Y.), the Strategic Priority Research Program of the Chinese Academy of Sciences (XDA16010500 Y.-G.Y.), Natural Science Foundation of China (92053115 Y.Y., 92153303 Y.Y., 91940304 Y.-G.Y.), the National Key R&D Program of China (2018YFA0801200 Y.Y.), the China Post-Doctoral Science Foundation (2022M713067 W.-L.Y.), CAS Key Research Projects of the Frontier Science (QYZDY-SSW-SMC027 Y.-G.Y.), the Youth Innovation Promotion Association of Chinese Academy of Sciences (CAS2018133 Y.Y.), the K. C. Wong Education Foundation, and Shanghai Municipal Science and Technology Major Project (2017SHZDZX01 Y.-G.Y.).

## Author contributions

Y.-G.Y. conceived this project, Y.-G.Y., P.-Y.Y., W.L., and Y.Y. supervised the study; W.-L.Y., W.-N.Q., and Z.-J.G. performed most of the experiments with assistance from H.-J.X.; T.Z. performed bioinformatics analysis; Q.Z., W.L., and K.X. performed the construction of *Nsun2* knock-out mice. B.S., Z.-Z.Y., and Y.Z. performed the construction of *Nsun2^{flox/flox}* mice. Y.-G.Y., Y.Y., W.-L.Y., W.-N.Q., T.Z., and Z.-J.G. discussed and integrated the data, Y.-G.Y., Y.Y., W.-L.Y., W.-N.Q., T.Z., Z.-J.G., and Y.-L.Z. wrote the manuscript with discussion and feedback from all authors.

## Competing interests

The authors declare no competing interests.
