## [Peer Review File · Nature Communications]

Nsun2 coupling with RoRyt shapes the fate of Th17 cells and promotes colitisREVIEWER COMMENTS

Reviewer #1 (Remarks to the Author):

The authors showed that Nsun2 is highly expressed in lymph node and spleen as well as in Th17 cells. Absence of Nsun2 resulted in lower expression of IL-17A and IL-17F. Interestingly, "Nsun2 did not affect RORgt expression in Th17 cells". However, the authors showed "a direct interaction between Nsun2 and RORgt" and they reveal that "Nsun2 couples with RORgt to control Th17 cell functional homeostasis through directly methylating Th17-specific transcripts, whereas the chromatin accessibility of RORgt and the overall transcription initiation were not affected by this process". In fact, "Nsum2 maintains Th17 cell homestasis through regulating the stability of IL-17A and IL-17F mRNAs dependent of m5C modification". In addition, the authors showed that targeting Nsun2 ameliorated Th17-mediated colitis in mice.

This is an in-depth and thorough study revealing the role of Nsun2 in the functional behavior of RORgt. The amount of data is enormous and the Supplementary data is essential to completely understand the performed research. The data are significant for the T cell biology / autoimmunity field. I have a few questions to elucidate some issues of the study.

1. Why did the authors use RPMI1610 in their Th17 differentiation studies, since it is known that this medium supports very low levels of Th17 polarization (please see: Veldhoen M, et al J Exp Med 2009, 206:43-9)?
2. Figure1a: please replace Fig. 1a by Extended Data Fig. 1a to show in the main manuscript the protein data and not the mRNA expression data. It will become then directly more clear that Nsun2 is predominantly expressed in lymph node and spleen.
3. The focus of Th17 molecules is very much on IL-17A and IL-17F, but does Nsun2 depletion/inhibition also influence other RORgt-mediated genes?
4. In line with question 3: IL-17A and IL-17F are not directly involved in Th17 polarization, does lack of Nsun2 (also) results in lower expression of IL-23R which may also lead to changes in pathogenic behavior of Th17 cells? Do the authors have data with respect to IL-23R expression?
5. Lack of Nsun2 resulted in amelioration of colitis. In terms of pathogenic versus non-pathogenic Th17 cells, did the authors had a change to examine IL-10 expression by these Th17 cells? Does lack of Nsun2 only result in lower IL-17A and IL-17F expression and therefore less interaction with cells in the environment as showed by the authors or (/and) did it also change the IL-17A/IL-10 ratio, meaning less pathogenic Th17 cells?
6. In terms of the idea of a therapeutic approach to neutralize the function of Nsun2, how specific is Nsun2 binding for RORgt? Please elaborate a bit more about this.

Minor points:

1. Introduction: Main, line 74: the authors mentioned "IL-17A/F". Please realize that IL-17A/F is a cytokine itself. If the authors mean IL-17A and IL-17F please mentioned them separately as IL-17A and IL-17F.
2. In the legend of Extended Data Fig 3, j,k,l is mentioned but the data is not shown. Please correct this.

Reviewer #2 (Remarks to the Author):

In this manuscript, Yang et al investigated the role of RNA 5-methylcytosine (m5C) methyltransferase Nsun2 in the regulation of Th17 cell fate and its function in colitis. They showed nicely Nsun2 mediating co-transcriptional m5C formation on Th17-related cytokine RNAs through directly interacting with Ror γ t, the key transcription factor for Th17 cells. They further showed that deficiency of Nsun2 inhibited colitis development in DSS model. Although the first part of data on Nsun2 regulating Th17 cell development through interacting with Ror γ t was quite convincing, the functional part in the regulation of colitis development was relatively weak, and more data are required to further confirm the conclusion. In addition, some concerns need to be addressed to further improve the quality of the manuscript.

- 1) Fig 4 and Extended Fig 11, although interesting, DSS-colitis model is an intestinal epithelial injure model which caused acute intestinal inflammation but not a T cell-mediated colitis model.

Thus it is not ideal for studying Th17 cells in this manuscript. At least a T cell-mediated colitis model, such as CD45RBhi T cell transfer model, should be used to confirm the role of Nsun2-regulated Th17 cells in colitis development.

2) Extended Fig 14, it is quite confusing that the authors showed that blockade of IL-17a/IL-17F inhibited DSS colitis as IL-17A has been shown by many groups as a protective cytokine of the intestines through enhancing intestinal barrier function and induction of antimicrobial peptide production to inhibit gut bacterial translocation. Again a T cell-mediated colitis model, such as CD45RBhi T cell transfer model, should be used to confirm those data. In addition, 4 groups of the mice should be used instead of 2 groups, i.e. WT with or without blockade of IL-17 and Nsun2 KO with or without blockade of IL-17.

3) Fig 4h, the immunofluorescence staining data is not convincing, FACS profiles of intestinal lamina propria T cells should be showed instead.

Reviewer #3 (Remarks to the Author):

In this manuscript by Yang and colleagues, the authors report the role of Nsun2 in the regulation of Th17 development and its pathogenic role in colitis. Topic is quite of interest and experimental setting is almost well done. However relevant points need to be addressed to improve the quality of the manuscript and solve some misleading reports.

- Extended fig 1c and d. Expression on Nsun2 in KO mice should be checked also in lymph node being on of the two tissue that express higher level of Nsun2 (together with spleen, already analysed)
- Values reported all over the text are frequencies and not ratios .
- The comparison between wt and Ko mice in terms of frequency of main subsets had to be done also in the lymph nodes (Extended fig 2b and d, fig 1d) in addition to spleen and PB. In particular lymph node analysis could allow to evaluate an high frequency of t cells from different subsets since the frequency of total T cells I higher than spleen and PB.
- Authors should comment the different expression of Nsun2 in Treg cells between mRNA and protein evaluation (fig 1b and Extended 3A). Moreover in Extended figure 3a, it is important to define if the different subsets were obtained as in figure legend 1b by in vitro stimulation or not.
- In extended fig 3f, the representative plots show a too much high level of "negative" population in particular for IL-4 and IFNg, leading to the possible exclusion of part of positive cells out of the top of the plot. Moreover also FOCP3 staining is not evaluable and not allow to define correctly positive and negative population. Data recovered from these kinds of setting plot could be not reliable.
- In extended figure 3 panels j-l described in the legend, were missing.
- Form which tissue T cells analysed in Extended fig 4d, were recovered: spleen? PB? Or lymph nodes?
- Axis of Figure 1i should be more detailed
- Extended fig 12 indicate IL-1 evaluation in the second column of representative dot plots. which phenotype was identified by expression of that cytokine?

Point-by-point responses to the reviewers' comments

1. Summary of major comments from the reviewers.

We have followed the insightful suggestions and comments from the reviewers to design and perform further experiments and analyses. We thank the editors and reviewers for the support and kind extension of our revision. The details of our revised results are summarized and listed in the following **Rebuttal Table 1**.

Rebuttal Table 1. Revision results for major comments from the reviewers.

	Question	Reviewers	New assays or New experiments
1	The focus of Th17 molecules is very much on IL-17A and IL-17F, but does Nsun2 depletion/inhibition also influence other RoR γ t-mediated genes, such as Il23r ?	1#: Q3-Q4	To systematically evaluate the influence of Nsun2 depletion on RoRγt targets, we have re-analyzed the mRNA-seq and RoRγt ChIP-seq data used in this study. We have identified that 41.2 % of down-regulated genes upon Nsun2 deletion were targeted by RoRγt, including Il17a, Il17f, Il1r1, Il1r2 and Il23r (Rebuttal Fig. 1a, b, Revised Supplementary Table 5). We also have applied the RT-qPCR to validate the expression level of these genes in wild-type and Nsun2^{CKO} Th17 cells, and found that the gene expression of Il1r2, Il23r, Il17a and Il17f was decreased upon Nsun2 deletion (Rebuttal Fig. 1c). As Il17a and Il17f were the most highly expressed genes in Th17 cells than the others (Rebuttal Fig. 1b, c), we mainly focused on IL-17A and IL-17F in the following functional analysis of Th17 cells. Although Il23r has been shown to be an upstream regulator of RoRγt and function in inducing Th17 cell differentiation¹, we didn't observe a changed expression of RoRγt upon Nsun2 deletion (original Fig. 1g and Extended Data Fig. 5e). Thus, it appears that RoRγt signaling is not regulated by the decreased Il23r expression under deficient Nsun2. In this study, we have focused on Th17-secreted IL-17A and IL-17F, the downstream of RoRγt, not only for their highly changed expression level upon Nsun2 deletion, but also for their critical roles in colitis development²⁻⁵.

2	Lack of Nsun2 resulted in amelioration of colitis. In terms of pathogenic versus non-pathogenic Th17 cells, did the authors had a change to examine IL-10 expression by these Th17 cells? Does lack of Nsun2 only result in lower IL-17A and IL-17F expression and therefore less interaction with cells in the environment as showed by the authors or (/and) did it also change the IL-17A/IL-10 ratio, meaning less pathogenic Th17 cells?	1#: Q5	Combined with mRNA-seq data and RT-qPCR experiment, we have found that, comparing to Il17a, Il10 expression was extremely low in Th17 cells (RPKM<1), and meanwhile, no significant change was observed upon Nsun2 depletion (Rebuttal Fig. 3). IL-10 had been shown to serve as an anti-inflammatory cytokine, endowing Treg cells with the ability to suppress pathogenic Th17 cell responses⁶. However, we failed to identify any effect of Nsun2-deletion on the vitro and in vivo Treg cell differentiation (Original Extended Data Fig. 3b-g and Revised Extended Data Fig. 3f-h). Additionally, under DSS-induced and T cell transfer-induced colitis models, we have not observed obvious changes in Treg cell frequency, but activity neutralization of IL-17A/both IL-17A and IL-17F by antibodies can rescue colitis development, suggesting that it is Nsun2-regulated IL-17A and IL-17F expression in Th17 cell, instead of IL-10 modulated Treg cells, participates in the ameliorated colitis pathogenesis caused by Nsun2 deletion (Rebuttal Fig. 5, Original Fig. 4g, Revised Fig. 4j, Revised Extended Data Fig. 13 and Fig. 14). In addition, RoRγt was not the transcription factor of Il10⁷ (Revised Supplementary Table 5). Instead, our data demonstrated that Nsun2 interacts with transcriptional factor RoRγt leading to the coupled m⁵C formation with transcription on Th17 cell-specific cytokine mRNAs, such as RoRγt targeted Il17a and Il17f genes. Collectively, we confirm that Nsun2-deficiency in Th17 cell results in lower IL-17A and IL-17F expression and less interaction with targeted cells in the environment, therefore reduces pathogenic Th17 cells.
---	--	--------	---

3	DSS-colitis model is an intestinal epithelial injury model which caused acute intestinal inflammation but not a T cell-mediated colitis model. Thus it is not ideal for studying Th17 cells in this manuscript. At least a T cell-mediated colitis model, such as CD45RB^{hi} T cell transfer model.	2#: Q1	We have established the CD45RB^{hi} T cell transfer model using Rag1^{-/-} mice. Through transferring CD4⁺CD25⁻CD45RB^{hi} naïve T cells into Rag1^{-/-} mice, we found that mice receiving Nsun2^{ckO} naïve T cells did not exhibit any signs of colitis up to 12 weeks after transfer, whereas mice receiving wild-type naïve T cells began losing weight in the sixth week after transfer (Rebuttal Fig. 4a, b, Revised Extended Data Fig. 13a, b). Nsun2^{ckO} recipients displayed smaller spleen sizes compared to wild-type recipients (Rebuttal Fig. 4c, Revised Extended Data Fig. 13c), and the colon length is normal similar with control group (PBS) at the 12th week after transfer (Rebuttal Fig. 4d, e, Revised Extended Data Fig. 13d, e). According to weight change, fecal viscosity, fecal occult blood, and athletic ability of recipients, we scored the recipient mice with colitis' DAI (Disease Activity Index), and found the colitis scores of Nsun2^{ckO} was significantly lower than wild-type naïve T cell recipients (Rebuttal Fig. 4f, Revised Extended Data Fig. 13f). Haematoxylin and eosin staining showed that Nsun2^{ckO} naïve T cells caused no T cell infiltration and inflammation, whereas wild-type T cells caused severe colonic inflammation and disrupted colon structure (Rebuttal Fig. 4g, Revised Extended Data Fig. 13g). Furthermore, FACS analysis discovered that Nsun2^{ckO} naïve T cells significantly reduced Th17 cell infiltration in lamina propria of intestinal in recipient mice compared to wild-type (Rebuttal Fig. 4h, Revised Extended Data Fig. 13h). Together, Nsun2^{ckO} naïve T cells do not promote disease in CD45RB^{hi} adoptive transfer colitis mouse model, which confirms the role of Nsun2-regulated Th17 cells in colitis development.
4	Extended Fig 14, it is quite confusing that the authors showed that blockade of IL-17a/IL-17F inhibited DSS colitis as IL-17A has been shown by many groups as a	2#: Q2	To define the function of IL-17A in colitis development using the CD45RB^{hi} T cell transfer model. Mice were injected intravenously with wild-type or Nsun2^{ckO} CD45RB^{hi} naïve T cells, followed by the treatment (two times per week) with neutralizing antibody of IL-17A alone or IL-17A plus IL-17F. The results showed that both treatments could lead to a total loss of colitis phenotype with normal colon macrograph and length in the mice receiving wild-type naïve T cells up to 12 weeks after transfer (Rebuttal

	protective cytokine of the intestines through enhancing intestinal barrier function and induction of antimicrobial peptide production to inhibit gut bacterial translocation. Again a T cell-mediated colitis model, such as CD45RB^{hi} T cell transfer model, should be used to confirm those data. In addition, 4 groups of the mice should be used instead of 2 groups, i.e. WT with or without blockade of IL-17 and Nsun2 KO with or without blockade of IL-17.		Fig. 5b, c, Revised Extended Data Fig. 14ab), which is similar as the findings in mice injected with Nsun2^{ckO} T cells. In contrast, mice receiving wild-type naïve T cells without antibody treatment began losing weight in the sixth week after transfer (Rebuttal Fig. 5a, Revised Extended Data Fig. 13a). Haematoxylin and eosin staining showed no T cell infiltration and inflammation after blockade of IL-17A alone or IL-17A plus IL-17F, consistent with the phenotype of Nsun2^{ckO} recipients (Rebuttal Fig. 4g, Rebuttal Fig. 5d, and Revised Extended Data Fig. 14c). Together, elimination of Th17 signaling in vivo diminished the development of colitis.
5	The immunofluorescence staining data is not convincing, FACS profiles of intestinal lamina propria T cells should be showed instead.	2#: Q3	We have optimized immunofluorescence staining on colon tissue of DSS-induced wild-type and Nsun2^{ckO} mice and statistical analysis of fluorescent density was also included. The result clearly confirmed that Th17 cell infiltration in colons of Nsun2^{ckO} mice significantly decreased compared to wild-type mice (Rebuttal Fig. 6a, b). Meanwhile, we have performed FACS profiles of intestinal lamina propria T cells from recipient mice transferred with the wild-type or Nsun2^{ckO} CD45RB^{hi} naïve T cells, and observed that the frequency of Th17 cells was significantly decreased in Nsun2^{ckO} T cell-transferred mice compared to wild-type T cell mice, while other lineage cells including Th1, Th2 and Treg cells showed minor or unaltered frequencies (Rebuttal Fig. 6c, and Revised Fig. 4j).

6	Expression on Nsun2 in KO mice should be checked also in lymph node being one of the two tissue that express higher level of Nsun2	3#: Q1	We have performed RT-qPCR and western blot assays to check the Nsun2 gene expression in lymph node and thymus of Nsun2 KO mice, and observed an extremely low mRNA and an undetectable protein expression in these two immune organs, validating the high efficiency of Nsun2 KO condition (Rebuttal Fig. 7a, b).
7	The comparison between WT and KO mice in terms of frequency of main subsets had to be done also in the lymph nodes.	3#: Q3	We have followed this suggestion to analyze the frequency of main T cell subsets in lymph nodes of wild-type and Nsun2^{-/-} mice by FACS. Consistent with the findings in spleen and PBMC, depletion of Nsun2 significantly dampened the percentage of Th17 cells in lymph nodes, while the changes in other subsets of T-cells were minor or unaltered (IFN-γ⁺ Th1 cells, IL-4⁺ Th2 cells, FOXP3⁺ Treg and IFN-γ⁺ CD8⁺ T cells) (Rebuttal Fig. 8 a, b). This data has now been included in the Revised Extended Data Fig. 3f-g.

REVIEWER COMMENTS

Reviewer #1 (Remarks to the Author):

The authors showed that *Nsun2* is highly expressed in lymph node and spleen as well as in Th17 cells. Absence of *Nsun2* resulted in lower expression of IL-17A and IL-17F. Interestingly, “*Nsun2* did not affect RoR γ t expression in Th17 cells”. However, the authors showed “a direct interaction between *Nsun2* and RoR γ t” and they reveal that “*Nsun2* couples with RoR γ t to control Th17 cell functional homeostasis through directly methylating Th17-specific transcripts, whereas the chromatin accessibility of RoR γ t and the overall transcription initiation were not affected by this process”. In fact, “*Nsun2* maintains Th17 cell homeostasis through regulating the stability of IL-17A and IL-17F mRNAs dependent of m⁵C modification”. In addition, the authors showed that targeting *Nsun2* ameliorated Th17-mediated colitis in mice.

This is an in-depth and thorough study revealing the role of *Nsun2* in the functional behavior of RoR γ t. The amount of data is enormous and the Supplementary data is essential to completely understand the performed research. The data are significant for the T cell biology / autoimmunity field. I have a few questions to elucidate some issues of the study.

We are very grateful to the reviewer for taking the time and effort to review the article and giving valuable comments. We have made the following point-by-point responses to answer the questions:

1. Why did the authors use RPMI1640 in their Th17 differentiation studies, since it is known that this medium supports very low levels of Th17 polarization (please see: Veldhoen M, et al *J Exp Med* 2009, 206:43-9)?

Response: Thanks for pointing out this. As a basic medium for T cell subset differentiation, many reports (>20) have utilized RPMI 1640 medium to culture the Th17 cells^{2,7-26}. We have listed the Th17 cell differentiation efficiency, ranging from 10 %-42.4 %, using RPMI 1640 medium in studies reported by Veldhoen M et al⁸. and others^{2,7-14} (**Rebuttal Table 1**). In our study, the Th17 differentiation had a comparable induction efficiency with around 45% (original **Figure 1e, f**). We have included the information of source companies and concentrations of our medium components in the corresponding Methods section (**Revised Manuscript Page 10 line 409-411 and Page 11 line 440-442**).

Rebuttal Table 1. The cell culture mediums of *in vitro* Th17 cell differentiation

	Composition of T cell culture medium	Differentiation efficiency of Th17 cell	Reference
Stockinger's lab	IMDM, 2 × 10 ⁻³ M l-glutamine, 100 U/ml penicillin, 100 µg/ml streptomycin, 5 × 10 ⁻⁵ M β-mercaptoethanol, and 5 % FCS.	About 54.8 %	Veldhoen M, et al. J Exp Med , 2009 ⁸ .
Stockinger's lab	RPMI 1640, 2 × 10 ⁻³ M l-glutamine, 100 U/ml penicillin, 100 µg/ml streptomycin, 5 × 10 ⁻⁵ M β-mercaptoethanol, and 5 % FCS.	42.4 %	Veldhoen M, et al. J Exp Med , 2009 ⁸ .

Huh's lab	RPMI1640, 10 % FBS, 25 mM glutamine, 55 μ M 2-mercaptoethanol, 100 U/ml penicillin, 100 mg/ml streptomycin	About 20 %	Hang, et al. Nature , 2019 ⁹ .
Lin's lab	RPMI 1640, 10 % FBS	About 16 %	Zhang, et al. Nature , 2020 ² .
Dong's lab	RPMI 1640, 100 U/mL of penicillin, 100 mg/mL streptomycin, 0.05 mM of β -mercaptoethanol and 10 % FBS	About 22 %	Wang et al. Immunity , 2020 ¹⁰ .
Wan's lab	RPMI 1640, 10 % FBS and 1 % antibiotics	About 20 %	Zhang et al. Nature , 2017 ¹¹ .
Müller's lab	RPMI 1640, 10 % FBS, and 1 % penicillin–streptomycin	About 10 %	Nicola et al. Nature , 2017 ¹² .
Gagliani's lab	RPMI 1640, 10 % FBS, 10 mM β -mercaptoethanol, Glutamax, 100 U/mL Penicillin/Streptomycin	About 40 %	Xu et al. Nature Communications , 2020 ¹³ .
Ding's lab	RPMI 1640, 10 % FBS (Invitrogen), penicillin-streptomycin (Invitrogen), 55 μ M β -mercaptoethanol, and 2 mM glutamine	About 35 %	Xu et al. Nature , 2017 ¹⁴ .
Yang's lab	RPMI1640, 10 % FBS (GIBCO), 1 % Penicilin-Streptomycin (GIBCO, 100 \times), 25 mM HEPES (BBI Life Sciences), non-essential amino acid (GIBCO, 100 \times) and 0.05 mM β -mercaptoethanol	About 45 %	Current work

2. *Figure 1a: please replace Fig. 1a by Extended Data Fig. 1a to show in the main manuscript the protein data and not the mRNA expression data. It will become then directly more clear that Nsun2 is predominantly expressed in lymph node and spleen.*

Response: Thanks for this suggestion. We have followed this advice and replaced the original **Fig. 1a** with the original **Extended Data Fig. 1a** in the revised manuscript (**Revised Fig. 1a and Extended Data Fig. 1a**).

3. *The focus of Th17 molecules is very much on IL-17A and IL-17F, but does Nsun2 depletion/inhibition also influence other RoR γ t-mediated genes?*

Response: Thanks for this thoughtful comment. To systematically evaluate the influence of *Nsun2* depletion on RoR γ t targets, we have re-analyzed the mRNA-seq and RoR γ t ChIP-seq data used in this study. We have identified that 41.2 % of down-regulated genes upon *Nsun2* deletion were targeted by RoR γ t, including *Il17a*, *Il17f*, *Il1r1*, *Il1r2* and *Il23r* (**Rebuttal Fig. 1a, b, Revised Supplementary Table 5**). We also have applied the RT-qPCR to validate the expression level of these genes in wild-type and *Nsun2*^{CKO} Th17 cells, and found that the gene expression of *Il1r2*, *Il23r*, *Il17a* and *Il17f* was decreased upon *Nsun2* deletion (**Rebuttal Fig. 1c**). As *Il17a* and *Il17f* were the most highly expressed genes in Th17 cells than the others (**Rebuttal Fig. 1b, c**), we mainly focused on IL-17A and IL-17F in the following functional analysis of Th17 cells.

Rebuttal Fig. 1 a, b, Venn diagram (**a**) and scatter plot (**b**) showing the overlapped genes between RoRyt targets and the down-regulated genes upon Nsun2 depletion. All data are representative of 2 independent experiments. **c**, RT-qPCR showing the mRNA level of *Il1r1*, *Il1r2*, *Il23r*, *Il17a* and *Il17f* in wild-type and *Nsun2*^{cKO} Th17 cells. *Gapdh* serves as an internal RNA control. All data are representative of 3 independent experiments. Data are two-tailed unpaired Student's *t* test. Error bars represent mean \pm s.e.m.

4. In line with question 3: IL-17A and IL-17F are not directly involved in Th17 polarization, does lack of Nsun2 (also) results in lower expression of IL-23R which may also lead to changes in pathogenic behavior of Th17 cells? Do the authors have data with respect to IL-23R expression?

Response: Thanks for the helpful comments. We have followed these suggestions to examine the mRNA expression of *Il23r* in the induced Th17 cells originated from wild-type, *Nsun2*^{-/-} or *Nsun2*^{cKO} mice, and found that Nsun2 loss could decrease the expression of *Il23r*, but both RNA and protein levels^{27,28} are relatively low, compared to *Il17a* and *Il17f* (**Rebuttal Fig. 1b,c**, **Rebuttal Fig. 2** and **Revised Supplementary Table 5**). Although *Il23r* has been shown to be an upstream regulator of RoRyt and function in inducing Th17 cell differentiation¹, we didn't observe a changed expression of RoRyt upon Nsun2 deletion (**Original Fig. 1g** and **Extended Data Fig. 5e**). Thus, it appears that RoRyt signaling is not regulated by the decreased *Il23r* expression under deficient Nsun2. In this study, we have focused on Th17-secreted IL-17A and IL-17F,

the downstream of RoR γ t, not only for their highly changed expression level upon *Nsun2* deletion, but also for their critical roles in colitis development²⁻⁵.

Rebuttal Fig. 2 RT-qPCR showing the mRNA expression of *Il23r* in wild-type and *Nsun2*^{-/-} Th17 cells. *Gapdh* serves as an internal RNA control. All data are representative of 3 independent experiments. Data are two-tailed unpaired Student's *t* test. Error bars represent mean \pm s.e.m.

5. Lack of *Nsun2* resulted in amelioration of colitis. In terms of pathogenic versus non-pathogenic Th17 cells, did the authors had a change to examine IL-10 expression by these Th17 cells? Does lack of *Nsun2* only result in lower IL-17A and IL-17F expression and therefore less interaction with cells in the environment as showed by the authors or (/and) did it also change the IL-17A/IL-10 ratio, meaning less pathogenic Th17 cells?

Response: Thanks for this comment. Combined with mRNA-seq data and RT-qPCR experiment, we have found that, comparing to *Il17a*, *Il10* expression was extremely low in Th17 cells (RPKM<1), and meanwhile, no significant change was observed upon *Nsun2* depletion (**Rebuttal Fig. 3**). IL-10 had been shown to serve as an anti-inflammatory cytokine, endowing Treg cells with the ability to suppress pathogenic Th17 cell responses⁶. However, we failed to identify any effect of *Nsun2*-deletion on the *vitro* and *in vivo* Treg cell differentiation (**Original Extended Data Fig. 3b-g and Revised Extended Data Fig. 3f-h**). Additionally, under DSS-induced and T cell transfer-induced colitis models, we have not observed obvious changes in Treg cell frequency, but activity neutralization of IL-17A/both IL-17A and IL-17F by antibodies can rescue colitis development, suggesting that it is *Nsun2*-regulated IL-17A and IL-17F expression in Th17 cell, instead of IL-10 modulated Treg cells, participates in the ameliorated colitis pathogenesis caused by *Nsun2* deletion (**Rebuttal Fig. 5, Original Fig. 4g, Revised Fig. 4j, Revised Extended Data Fig. 13 and Fig. 14**). In addition, RoR γ t was not the transcription factor of *Il10*⁷ (**Revised Supplementary Table 5**). Instead, our data demonstrated that *Nsun2* interacts with transcriptional factor RoR γ t leading to the coupled m⁵C formation with transcription on Th17 cell-specific cytokine mRNAs, such as RoR γ t targeted *Il17a* and *Il17f* genes. Collectively, we confirm that *Nsun2*-deficiency in Th17 cell results in lower IL-17A and IL-17F expression and less

interaction with targeted cells in the environment, therefore reduces pathogenic Th17 cells.

Rebuttal Fig. 3 The gene expression changes of *Il10* and *Il17a* upon *Nsun2* deletion in Th17 cells. **a, b**, The RPKM (Reads Per Kilobase per Million mapped reads) showing the mRNA expression of *Il10* and *Il17a* in wild-type and *Nsun2^{cKO}* Th17 cells. **c**, RT-qPCR showing the gene expression of *Il10* and *Il17a* in wild-type and *Nsun2^{cKO}* Th17 cells. All data are representative of 2 independent experiments. Data are two-tailed unpaired Student's *t* test. Error bars represent mean \pm s.e.m.

6. In terms of the idea of a therapeutic approach to neutralize the function of *Nsun2*, how specific is *Nsun2* binding for RoR γ t? Please elaborate a bit more about this.

Response: Thanks for pointing out this. Our findings from the *Nsun2* IP-MS and *in vivo* and *in vitro* protein-protein interaction experiments demonstrated that *Nsun2* can directly interact with RoR γ t, and the enzymatic activity site of *Nsun2* is critical for its regulated Th17 cell function (**Original Fig. 2a-d, Extended Data Fig. 6a, b, Fig. 3c-f and Fig. 10a**). More importantly, using a CD45RB^{hi} adoptive transfer mouse colitis model, we found that through directly blocking the RoR γ t downstream signaling of IL-17A and IL-17F by neutralizing antibodies, mice receiving wild-type CD45RB^{hi} naïve T cells were protected from colitis development with same phenotypes as *Nsun2^{cKO}* CD45RB^{hi} naïve T cell transferred mice (**Revised Extended Data Fig. 13 and Fig. 14**). Thus, these two molecules as the co-targets of *Nsun2* and RoR γ t have the potential to be used as the therapeutic targets for colitis treatment.

Minor points:

1. Introduction: Main, line 74: the authors mentioned "IL-17A/F". Please realize that IL-17A/F is a cytokine itself. If the authors mean IL-17A and IL-17F please mentioned them separately as IL-17A and IL-17F.

Response: Thanks for this suggestion. We have checked the full text and corrected IL-17A/F to IL-17A and IL-17F in the **Revised Manuscript Page 3 line 78**.

2. In the legend of Extended Data Fig 3 j,k,l is mentioned but the data is not shown. Please correct this.

Response: Thanks for pointing out this. We have deleted the legend information of original **Extended Data Fig. 3j-l** and corrected it in the revised manuscript.

Reviewer #2 (Remarks to the Author):

In this manuscript, Yang et al investigated the role of RNA 5-methylcytosine (m^5C) methyltransferase Nsun2 in the regulation of Th17 cell fate and its function in colitis. They showed nicely Nsun2 mediating co-transcriptional m^5C formation on Th17-related cytokine RNAs through directly interacting with RoR γ t, the key transcription factor for Th17 cells. They further showed that deficiency of Nsun2 inhibited colitis development in DSS model. Although the first part of data on Nsun2 regulating Th17 cell development through interacting with RoR γ t was quite convincing, the functional part in the regulation of colitis development was relatively weak, and more data are required to further confirm the conclusion. In addition, some concerns need to be addressed to further improve the quality of the manuscript.

Thanks to the reviewer for taking the time and efforts to review our articles and putting forward constructive suggestions. We have made the following point-by-point responses to answer the questions:

1. Fig 4 and Extended Fig 11, although interesting, DSS-colitis model is an intestinal epithelial injure model which caused acute intestinal inflammation but not a T cell-mediated colitis model. Thus it is not ideal for studying Th17 cells in this manuscript. At least a T cell-mediated colitis model, such as CD45RB^{hi} T cell transfer model, should be used to confirm the role of Nsun2-regulated Th17 cells in colitis development.

Response: Thanks for this very thoughtful suggestion. Although DSS-induced colitis has been widely used in Th17 cell regulating intestinal homeostasis ^{2,29-49}, we agree with this reviewer that CD45RB^{hi} T cell transfer model is more suitable to study T cell-mediated colitis.

We have followed this insightful suggestion to establish the CD45RB^{hi} T cell transfer model using *Rag1*^{-/-} mice. Through transferring CD4⁺CD25⁻CD45RB^{hi} naïve T cells into *Rag1*^{-/-} mice, we found that mice receiving *Nsun2*^{ckO} naïve T cells did not exhibit any signs of colitis up to 12 weeks after transfer, whereas mice receiving wild-type naïve T cells began losing weight in the sixth week after transfer (**Rebuttal Fig. 4a, b, Revised Extended Data Fig. 13a, b**). *Nsun2*^{ckO} recipients displayed smaller spleen sizes compared to wild-type recipients (**Rebuttal Fig. 4c, Revised Extended Data Fig. 13c**), and the colon length is normal similar with control group (PBS) at the 12th week after transfer (**Rebuttal Fig. 4d, e, Revised Extended Data Fig. 13d, e**). According to weight change, fecal viscosity, fecal occult blood, and athletic ability of recipients, we scored the recipient mice with colitis' DAI (Disease Activity Index), and found the colitis scores of *Nsun2*^{ckO} was significantly lower than wild-type naïve T cell recipients (**Rebuttal Fig. 4f, Revised Extended Data Fig. 13f**). Haematoxylin and eosin staining showed that *Nsun2*^{ckO} naïve T cells caused no T cell infiltration and inflammation, whereas wild-type T cells caused severe colonic inflammation and disrupted colon structure (**Rebuttal Fig. 4g, Revised Extended Data Fig. 13g**).

Furthermore, FACS analysis discovered that *Nsun2^{ckO}* naïve T cells significantly reduced Th17 cell infiltration in lamina propria of intestinal in recipient mice compared to wild-type (**Rebuttal Fig. 4h, Revised Extended Data Fig. 13h**). Together, *Nsun2^{ckO}* naïve T cells do not promote disease in CD45RB^{hi} adoptive transfer colitis mouse model, which confirms the role of Nsun2-regulated Th17 cells in colitis development.

we have included these new results in revised manuscript page 7 line 252-262 and Revised Extended Data Fig. 13, and the subsequent figure panel numbers are changed accordingly. Meanwhile, the detailed experimental methods for this experiment have been added in revised manuscript Methods section in page 15 line 602-609.

Rebuttal Fig. 4 a-h, The phenotype of PBS ($n = 5$), wild-type ($n = 5$) and *Nsun2^{ckO}* ($n = 5$) naïve T cell adoptively transferring into *Rag1^{-/-}* recipient mice. Body weight changes (**a**), representative image (**b**), representative image of spleen (**c**), representative macrograph of colons (**d**), the colon length (**e**), colitis score (**f**), representative hematoxylin-eosin (H&E) staining indicated the severity of colons damage and inflammatory infiltration (**g**), flow cytometry of CD4⁺IL-17A⁺IFN γ ⁺, CD4⁺IL-17A⁺IFN γ ⁺ and CD4⁺IL-17A⁺IFN γ ⁺ cells in intestinal lamina propria T cells of wild-type and *Nsun2^{ckO}* naïve T cell recipient mice (**h**). PBS vehicle injection was used as

control.. n = number of biological replicates. Data are two-tailed unpaired Student's t test. Error bars represent mean \pm s.e.m (**a, f, g, h**).

2. *Extended Fig 14, it is quite confusing that the authors showed that blockade of IL-17a/IL-17F inhibited DSS colitis as IL-17A has been shown by many groups as a protective cytokine of the intestines through enhancing intestinal barrier function and induction of antimicrobial peptide production to inhibit gut bacterial translocation. Again a T cell-mediated colitis model, such as CD45RB^{hi} T cell transfer model, should be used to confirm those data. In addition, 4 groups of the mice should be used instead of 2 groups, i.e. WT with or without blockade of IL-17 and Nsun2 KO with or without blockade of IL-17.*

Response: Thanks for pointing out this. As mentioned by this reviewer, IL-17A has been shown to be protective cytokine of the intestine through inducing antimicrobial peptide production. However, the precise function of IL-17A in inflammatory bowel disease (IBD) remains unclear. Pedersen et al reported that Th17 cell development correlates with colitis progression, and concurrent neutralization of their cytokine products of IL-17A and IL-17F ameliorates intestinal inflammation, whereas neutralization of IL-17A or IL-17F alone was inefficient⁵. Dong et al reported that in DSS- induced acute experimental colitis, IL-17A knockout increased the colon damage, but IL-17F deficiency resulted in reduced colitis development⁴. But other studies have showed that IL-17A, a representative cytokine produced by Th17 cells, has an important role for the pathological process of inflammatory diseases^{2,3}. Mazda et al. investigated the function of IL-17A in IBD and found that *Il17a* deficient mice showed faint manifestations of colitis with 90 % of the survival mice over the DSS treatment, compared with wild-type mice. They further observed that histological change, epithelial damage and inflammatory infiltrates were not obvious upon *Il17a* deletion³. In addition, Lin et al. showed that *Il17a* and *Rorc* expression has positive correlation with colitis development in human patients; moreover, they demonstrated that IL-17⁺ Th17 cell is the main driver of DSS-induced colitis². Overall, the precise function of IL-17A needs to be explored further.

We have followed this reviewer's concern to define the function of IL-17A in colitis development using the CD45RB^{hi} T cell transfer model. Mice were injected intravenously with wild-type or *Nsun2^{ckO}* CD45RB^{hi} naïve T cells, followed by the treatment (two times per week) with neutralizing antibody of IL-17A alone or IL-17A plus IL-17F. The results showed that both treatments could alleviate colitis phenotype with normal colon macrograph and length in the mice receiving wild-type naïve T cells up to 12 weeks after transfer (**Rebuttal Fig. 5b, c, Revised Extended Data Fig. 14ab**), which is similar as the findings in mice injected with *Nsun2^{ckO}* T cells (**Rebuttal Fig. 4d, e and Revised Extended Data Fig. 13d, e**). In contrast, mice receiving wild-type naïve T cells without antibody treatment began losing weight in the sixth week after transfer (**Rebuttal Fig. 5a, Revised Extended Data Fig. 13a**). Haematoxylin and eosin staining showed no T cell infiltration and inflammation after blockade of IL-17A alone or IL-17A plus IL-17F, consistent with the phenotype of *Nsun2^{ckO}* recipients (**Rebuttal**

Fig. 4g and 5d, and Revised Extended Data Fig. 13g and Fig. 14c). Together, elimination of Th17 signaling *in vivo* diminished the development of colitis.

We have included these new results by replacing the **original manuscript line 274-277 and Extended Data Fig. 14a-d** obtained from the DSS-induced colitis model with the new results from the CD45RB^{hi} T cell transfer models **in this revised manuscript (page 7 line 262-268 and revised Extended Data Fig. 14a-c)**. Meanwhile, the detailed experimental methods for this experiment have been included in the Methods section in revised manuscript page 15 line 602-609.

Rebuttal Fig. 5 Blockage of IL-17A or both IL-17A and IL-17F signaling eliminated colitis development in CD4⁺ CD25⁻ CD45RB^{hi} adoptive transfer colitis mouse model. **a-d**, The phenotype of *Rag1^{-/-}* host mice transferred with wild-type and *Nsun2^{cKO}* naïve CD45RB^{hi} T cell, followed by the treatment (two times per week) with neutralizing

antibody of IL-17A alone or IL-17A plus IL-17F. Body weight changes (a), the representative macrograph of colons (b), colon length (c), and hematoxylin-eosin (H&E) staining indicated the severity of colons damage and inflammatory infiltration (d). PBS vehicle injection was used as control. n = number of biological replicates. Data are two-tailed unpaired Student's t test. Error bars represent mean \pm s.e.m.

3. Fig 4h, the immunofluorescence staining data is not convincing, FACS profiles of intestinal lamina propria T cells should be showed instead.

Response: Thanks for this insightful comment. We have optimized immunofluorescence staining on colon tissue of DSS-induced wild-type and *Nsun2^{cKO}* mice and statistical analysis of fluorescent density was also included. The result clearly confirmed that Th17 cell infiltration in colons of *Nsun2^{cKO}* mice significantly decreased compared to wild-type mice (**Rebuttal Fig. 6a, b**). We have now replaced the original Fig. 4h with the new results in **Revised Fig. 4h** and **Fig. 4i**.

Meanwhile, we have performed FACS profiles of intestinal lamina propria T cells from recipient mice transferred with the wild-type or *Nsun2^{cKO}* CD45RB^{hi} naïve T cells, and observed that the frequency of Th17 cells was significantly decreased in *Nsun2^{cKO}* T cell-transferred mice compared to wild-type T cell mice, while other lineage cells including Th1, Th2 and Treg cells showed minor or unaltered frequencies (**Rebuttal Fig. 6c**, and **Revised Fig. 4j**). The subsequent figure panel numbers were changed accordingly.

Rebuttal Fig. 6 a, b, Immunofluorescence staining assay showing Th17 infiltration in the colons of DSS-induced wild-type and *Nsun2^{cKO}* mice, scale bar, 100 μ m. Fluorescence signal statistics (**b**), wild-type ($n = 18$), *Nsun2^{cKO}* ($n = 25$). **c**, FACS showing the frequencies of T cell subsets in intestinal lamina propria T cells in recipient

mice of transferring wild-type and *Nsun2*^{ckO} CD4⁺ CD25⁻ CD45RB^{hi} naïve T cell for 12 weeks, including Th1 cell, Th2 cell, Th17 cell and Treg cell. All data are representative of 3 independent experiments. Data are two-tailed unpaired Student's *t* test. Error bars represent mean ± s.e.m.

Reviewer #3 (Remarks to the Author):

In this manuscript by Yang and colleagues, the authors report the role of Nsun2 in the regulation of Th17 development and its pathogenic role in colitis. Topic is quite of interest and experimental setting is almost well done. However relevant points need to be addressed to improve the quality of the manuscript and solve some misleading reports.

We are very grateful to the reviewers for their valuable suggestions which are highly helpful in improving the quality of our manuscript. We have made the following point-by-point responses to answer the questions:

1. Extended fig 1c and d. Expression on Nsun2 in KO mice should be checked also in lymph node being one of the two tissue that express higher level of Nsun2 (together with spleen, already analyzed)

Response: Thanks for this thoughtful suggestion. We have performed RT-qPCR and western blot assays to check the *Nsun2* gene expression in lymph node and thymus of *Nsun2* KO mice, and observed an extremely low mRNA and an undetectable protein expression in these two immune organs, validating the high efficiency of *Nsun2* KO condition (**Rebuttal Fig. 7a, b**). This data is now included in the **Revised Extended Data Fig. 1c-d**).

Rebuttal Fig. 7 RT-qPCR (a) and Western blot (b) showing the knockout efficiency of *Nsun2* in lymph node and thymus in wild-type and *Nsun2*^{-/-} mice. All data are

representative of 3 independent experiments. Data are two-tailed unpaired Student's *t* test. Error bars represent mean \pm s.e.m.

2. Values reported all over the text are frequencies and not ratios.

Response: Thanks for pointing out this. We have read the full text, and replaced T cell ratios with T cell frequencies in this revised manuscript and figures (**Revised manuscript page 3 line 96,98, and 106, page 7 line 246 and 247, page 17 line 723 and 725, page 21 line 864 and 868, page 23 line 909, page 24 line 925; Revised Supplementary page 3 line 21 and 23, page 5 line 39; Revised Fig. 1d, f, i, and Extended Fig. 2 a and c, Fig. 3b-e).**

3. The comparison between *wt* and *Ko* mice in terms of frequency of main subsets had to be done also in the lymph nodes (Extended fig 2b and d, fig 1d) in addition to spleen and PB. In particular lymph node analysis could allow to evaluate a high frequency of *t* cells from different subsets since the frequency of total T cells is higher than spleen and PB.

Response: Thanks for this valuable comment. We have followed this suggestion to analyze the frequency of main T cell subsets in lymph nodes of wild-type and *Nsun2*^{-/-} mice by FACS. Consistent with the findings in spleen and PBMC, depletion of *Nsun2* significantly dampened the percentage of Th17 cells in lymph nodes, while the changes in other subsets of T-cells were minor or unaltered (IFN- γ ⁺ Th1 cells, IL-4⁺ Th2 cells, FOXP3⁺ Treg and IFN- γ ⁺ CD8⁺ T cells) (**Rebuttal Fig. 8a, b**). This data has now been included in the **Revised Extended Data Fig. 3f, g**, and the subsequent figure panel numbers were changed accordingly.

Rebuttal Fig. 8 FACS analysis showing the percentage of CD4⁺ IFN- γ ⁺ Th1 cells, CD4⁺ IL-4⁺ Th2 cells, CD4⁺ IL-17A⁺ Th17 cells, CD4⁺ FOXP3⁺ Treg cells and CD8⁺ IFN- γ ⁺ cells in lymph node. Sorted lymphocyte from lymph nodes of littermate wild-type and *Nsun2*^{-/-} mice following stimulation with PMA (100 ng/ml) / ionomycin (500 ng/ml) for 5.5 h in the presence of Brefeldin A (1000 \times) were analyzed by FACS. All data are representative of 3 independent experiments. Data are two-tailed unpaired Student's *t* test. Error bars represent mean \pm s.e.m.

4. Authors should comment the different expression of *Nsun2* in Treg cells between mRNA and protein evaluation (fig 1b and Extended 3a). Moreover, in Extended figure 3a, it is important to define if the different subsets were obtained as in figure legend 1b by *in vitro* stimulation or not.

Response: We thank the reviewer to point out this. The inconsistent gene expression between mRNA and protein levels does exist in the particular locus or the same cell type. In one review article published in *Cell* by Aebersold et al.⁵⁰ have summarized the possible factors based on the published data which might influence the protein expression independent of mRNA levels, including (1) mRNA sequence that affects the mRNA translation efficiency (eg, gene's codon composition, upstream open reading frames (uORFs), internal ribosome entry sites); (2) the binding of proteins to regulatory elements on the transcript, the binding of non-coding RNAs such as micro-RNAs or the relative availability of transcript and (charged) ribosomes can modulate the gene translation rates; (3) the complex ubiquitin-proteasome pathway or autophagy may regulate proteins degradation independent transcript concentrations; (4) protein synthesis delay: transcript level is probably changed during protein synthesis; (5) protein transport: protein export spatially disconnects proteins from the transcripts they were synthesized from. In addition, recent findings also demonstrate that mRNA methylation (eg, m⁶A and m⁵C) also play a critical role in regulating translation efficiency⁵¹⁻⁵⁴. All these factors might individually or cooperatively contribute to the mRNA translation efficiency leading to differential abundance in mRNA and protein expression for one gene in the same cell type, which could also reasonably explain our findings in Treg cells showing different levels of *Nsun2* mRNA and protein expression.

The cell samples showed in original **Figure 1b** and **Extended Data Fig. 3a** were obtained from the *in vitro* stimulation. We have added the sample acquisition information in the revised manuscript (**Supplementary Information page 4, line 34-36**).

5. In extended fig 3f, the representative plots show a too much high level of “negative” population in particular for IL-4 and IFN γ , leading to the possible exclusion of part of positive cells out of the top of the plot. Moreover, also FOXP3 staining is not evaluable and not allow to define correctly positive and negative population. Data recovered from these kinds of setting plot could be not reliable.

Response: We thank the reviewer to point out this issue. There are multiple previously reported studies presenting FoxP3 staining based on FoxP3-pacific-blue (Biolegend, Cat:126409) antibody⁵⁵⁻⁵⁸. The same antibody was also employed in our staining analysis. Besides, we have followed the advice from this reviewer to re-analyze the FACS data of Treg from *in vitro* differentiation by FlowJo. (**Rebuttal Fig. 9**). This data is included in the revised **Extended Data Fig. 3h** to replace the original **Extended Data Fig. 3f**.

Rebuttal Fig. 9 FACS assay showing the frequencies of *in vitro* differentiation of CD4⁺ IFN-γ⁺ Th1, CD4⁺ IL-4⁺ Th2, CD4⁺ IL-17A⁺ Th17, CD4⁺ FOXP3⁺ Treg and CD8⁺ IFN-γ⁺ cells in littermate wild-type and *Nsun2*^{-/-} mice.

6. In extended figure 3 panels j-l described in the legend, were missing.

Response: Thanks for pointing out this. We have deleted the legend information of original **Extended Data Fig. 3j-l** and corrected it in the revised manuscript.

7. Form which tissue T cells analyzed in Extended fig 4d, were recovered: spleen? PB? Or lymph nodes?

Response: Thanks for this insightful comment. The T cells showed in Extended Fig 4d were sorted from lymph nodes. We have added this information to figure legend in revised **Extended Data Fig. 4d (Revised Supplementary Information, page 6, line 60-62)**.

7. Axis of Figure 1i should be more detailed.

Response: Thanks for this suggestion. We have modified the axis of the **Figure 1i** in revised manuscript and marked with red.

8. Extended fig 12 indicate IL-1 evaluation in the second column of representative dot plots. which phenotype was identified by expression of that cytokine?

Response: Thank you for pointing out this. We have corrected IL-1 in the **Revised Extended Fig. 12** to IL-4, and marked with red.

Supplementary notes:

1. For half-life of minigenes Il17a-C (with unmodified C) and Il17a-m⁵C, the three time points (0, 20, 40 min) was collected, therefore we have modified in **Revised manuscript page 6 line 228**, and the detailed experimental methods for this experiment have been added in **revised manuscript page 14 line 585-593**.

2. In order to better describe experimental methods, we have included several key references to the Method section and marked with red (**Revised manuscript page 11 line 427-428, page 13 line 513, page 14 line 580, page 15 line 596, page 16 line 677**). The reference codes are changed accordingly.

References:

- 1 Zhou, L. *et al.* IL-6 programs T_H-17 cell differentiation by promoting sequential engagement of the IL-21 and IL-23 pathways. *Nat Immunol* **8**, 967-974, doi:10.1038/ni1488 (2007).
- 2 Zhang, M. *et al.* A STAT3 palmitoylation cycle promotes T_H17 differentiation and colitis. *Nature* **586**, 434-439, doi:10.1038/s41586-020-2799-2 (2020).
- 3 Ito, R. *et al.* Involvement of IL-17A in the pathogenesis of DSS-induced colitis in mice. *Biochem Biophys Res Commun* **377**, 12-16, doi:10.1016/j.bbrc.2008.09.019 (2008).
- 4 Yang, X. O. *et al.* Regulation of inflammatory responses by IL-17F. *J Exp Med* **205**, 1063-1075, doi:10.1084/jem.20071978 (2008).
- 5 Wedeby Schmidt, E. G. *et al.* T_H17 cell induction and effects of IL-17A and IL-17F blockade in experimental colitis. *Inflamm Bowel Dis* **19**, 1567-1576, doi:10.1097/MIB.0b013e318286fa1c (2013).
- 6 Chaudhry, A. *et al.* Interleukin-10 signaling in regulatory T cells is required for suppression of Th17 cell-mediated inflammation. *Immunity* **34**, 566-578, doi:10.1016/j.immuni.2011.03.018 (2011).
- 7 Ciofani, M. *et al.* A validated regulatory network for Th17 cell specification. *Cell* **151**, 289-303, doi:10.1016/j.cell.2012.09.016 (2012).
- 8 Veldhoen, M., Hirota, K., Christensen, J., O'Garra, A. & Stockinger, B. Natural agonists for aryl hydrocarbon receptor in culture medium are essential for optimal differentiation of Th17 T cells. *J Exp Med* **206**, 43-49, doi:10.1084/jem.20081438 (2009).
- 9 Hang, S. *et al.* Bile acid metabolites control T_H17 and T_{reg} cell differentiation. *Nature* **576**, 143-148, doi:10.1038/s41586-019-1785-z (2019).
- 10 Wang, X. *et al.* Febrile temperature critically controls the differentiation and pathogenicity of T helper 17 cells. *Immunity* **52**, 328-341 e325, doi:10.1016/j.immuni.2020.01.006 (2020).
- 11 Zhang, S. *et al.* Reversing SKI-SMAD4-mediated suppression is essential for T_H17 cell differentiation. *Nature* **551**, 105-109, doi:10.1038/nature24283 (2017).
- 12 Wilck, N. *et al.* Salt-responsive gut commensal modulates T_H17 axis and disease. *Nature* **551**, 585-589, doi:10.1038/nature24628 (2017).
- 13 Xu, H. *et al.* The induction and function of the anti-inflammatory fate of T_H17 cells. *Nat Commun* **11**, 3334, doi:10.1038/s41467-020-17097-5 (2020).
- 14 Xu, T. *et al.* Metabolic control of T_H17 and induced T_{reg} cell balance by an epigenetic mechanism. *Nature* **548**, 228-233, doi:10.1038/nature23475 (2017).
- 15 Liu, Y. *et al.* Heme oxygenase-1 restores impaired GARP⁺CD4⁺CD25⁺ regulatory T cells from patients with acute coronary syndrome by upregulating LAP and GARP expression on activated T lymphocytes. *Cell Physiol Biochem* **35**, 553-570, doi:10.1159/000369719 (2015).
- 16 Pai, C., Walsh, C. M. & Fruman, D. A. Context-specific function of S6K2 in Th cell differentiation. *J Immunol* **197**, 3049-3058, doi:10.4049/jimmunol.1600167

- (2016).
- 17 Kwon, M. J., Ma, J., Ding, Y., Wang, R. & Sun, Z. Protein kinase C-theta promotes Th17 differentiation via upregulation of Stat3. *J Immunol* **188**, 5887-5897, doi:10.4049/jimmunol.1102941 (2012).
- 18 Zhang, Y. *et al.* MKP-1 is necessary for T cell activation and function. *J Biol Chem* **284**, 30815-30824, doi:10.1074/jbc.M109.052472 (2009).
- 19 Mangalam, A. K. *et al.* AMP-activated protein kinase suppresses autoimmune central nervous system disease by regulating M1-type macrophage-Th17 axis. *J Immunol* **197**, 747-760, doi:10.4049/jimmunol.1501549 (2016).
- 20 Zhang, M. A. *et al.* Antagonizing peroxisome proliferator-activated receptor alpha activity selectively enhances Th1 immunity in male mice. *J Immunol* **195**, 5189-5202, doi:10.4049/jimmunol.1500449 (2015).
- 21 Mitsuki, Y. Y., Tuen, M. & Hioe, C. E. Differential effects of HIV transmission from monocyte-derived dendritic cells vs. monocytes to IL-17⁺CD4⁺ T cells. *J Leukoc Biol* **101**, 339-350, doi:10.1189/jlb.4A0516-216R (2017).
- 22 Nakaya, M. *et al.* Inflammatory T cell responses rely on amino acid transporter ASCT2 facilitation of glutamine uptake and mTORC1 kinase activation. *Immunity* **40**, 692-705, doi:10.1016/j.immuni.2014.04.007 (2014).
- 23 Wu, L. *et al.* Alteration of Th17 and Treg cells in patients with unexplained recurrent spontaneous abortion before and after lymphocyte immunization therapy. *Reprod Biol Endocrinol* **12**, 74, doi:10.1186/1477-7827-12-74 (2014).
- 24 Chen, R. Y. *et al.* Estradiol inhibits Th17 cell differentiation through inhibition of ROR γ T transcription by recruiting the ERalpha/REA complex to estrogen response elements of the ROR γ T promoter. *J Immunol* **194**, 4019-4028, doi:10.4049/jimmunol.1400806 (2015).
- 25 Manel, N., Unutmaz, D. & Littman, D. R. The differentiation of human T_H-17 cells requires transforming growth factor- β and induction of the nuclear receptor ROR γ t. *Nat Immunol* **9**, 641-649, doi:10.1038/ni.1610 (2008).
- 26 Liu, H. *et al.* ERK differentially regulates Th17- and Treg-cell development and contributes to the pathogenesis of colitis. *Eur J Immunol* **43**, 1716-1726, doi:10.1002/eji.201242889 (2013).
- 27 Wu, C. *et al.* Induction of pathogenic T_H17 cells by inducible salt-sensing kinase SGK1. *Nature* **496**, 513-517, doi:10.1038/nature11984 (2013).
- 28 Meyer Zu Horste, G. *et al.* RBPJ controls development of pathogenic Th17 cells by regulating IL-23 receptor expression. *Cell Rep* **16**, 392-404, doi:10.1016/j.celrep.2016.05.088 (2016).
- 29 Schnell, A. *et al.* Stem-like intestinal Th17 cells give rise to pathogenic effector T cells during autoimmunity. *Cell* **184**, 6281-6298 e6223, doi:10.1016/j.cell.2021.11.018 (2021).
- 30 Alexander, M. *et al.* Human gut bacterial metabolism drives Th17 activation and colitis. *Cell Host Microbe* **30**, 17-30 e19, doi:10.1016/j.chom.2021.11.001 (2022).
- 31 Saleh, M. M. *et al.* Colitis-induced Th17 cells increase the risk for severe subsequent clostridium difficile infection. *Cell Host Microbe* **25**, 756-765 e755,

- doi:10.1016/j.chom.2019.03.003 (2019).
- 32 Perez, L. G. *et al.* TGF- β signaling in Th17 cells promotes IL-22 production and colitis-associated colon cancer. *Nat Commun* **11**, 2608, doi:10.1038/s41467-020-16363-w (2020).
- 33 Xu, X. *et al.* Madecassic acid, the contributor to the anti-colitis effect of madecassoside, enhances the shift of Th17 toward Treg cells via the PPAR γ /AMPK/ACC1 pathway. *Cell Death Dis* **8**, e2723, doi:10.1038/cddis.2017.150 (2017).
- 34 Schmidt, N. *et al.* Targeting the proteasome: partial inhibition of the proteasome by bortezomib or deletion of the immunosubunit LMP7 attenuates experimental colitis. *Gut* **59**, 896-906, doi:10.1136/gut.2009.203554 (2010).
- 35 Wang, J. *et al.* Larval *Echinococcus multilocularis* infection reduces dextran sulphate sodium-induced colitis in mice by attenuating T helper type 1/type 17-mediated immune reactions. *Immunology* **154**, 76-88, doi:10.1111/imm.12860 (2018).
- 36 Acharya, S. *et al.* Amelioration of experimental autoimmune encephalomyelitis and DSS induced colitis by NTG-A-009 through the inhibition of Th1 and Th17 cells differentiation. *Sci Rep* **8**, 7799, doi:10.1038/s41598-018-26088-y (2018).
- 37 Feng, T. *et al.* Th17 cells induce colitis and promote Th1 cell responses through IL-17 induction of innate IL-12 and IL-23 production. *J Immunol* **186**, 6313-6318, doi:10.4049/jimmunol.1001454 (2011).
- 38 Zhu, J. *et al.* IL-33 alleviates DSS-induced chronic colitis in C57BL/6 mice colon lamina propria by suppressing Th17 cell response as well as Th1 cell response. *Int Immunopharmacol* **29**, 846-853, doi:10.1016/j.intimp.2015.08.032 (2015).
- 39 Sheng, Y. *et al.* The effect of 6-gingerol on inflammatory response and Th17/Treg balance in DSS-induced ulcerative colitis mice. *Ann Transl Med* **8**, 442, doi:10.21037/atm.2020.03.141 (2020).
- 40 Longhi, M. S. *et al.* Bilirubin suppresses Th17 immunity in colitis by upregulating CD39. *JCI Insight* **2**, doi:10.1172/jci.insight.92791 (2017).
- 41 Yang, F. *et al.* Th1/Th2 balance and Th17/Treg-mediated immunity in relation to murine resistance to dextran sulfate-induced colitis. *J Immunol Res* **2017**, 7047201, doi:10.1155/2017/7047201 (2017).
- 42 Monk, J. M. *et al.* Th17 cell accumulation is decreased during chronic experimental colitis by (n-3) PUFA in Fat-1 mice. *J Nutr* **142**, 117-124, doi:10.3945/jn.111.147058 (2012).
- 43 Ho, C. C. *et al.* Transcriptional interactomic inhibition of ROR α suppresses Th17-related inflammation. *J Inflamm Res* **14**, 7091-7105, doi:10.2147/JIR.S344031 (2021).
- 44 Lai, H. *et al.* Root extract of *Lindera aggregata* (Sims) Kosterm. modulates the Th17/Treg balance to attenuate DSS-Induced colitis in mice by IL-6/STAT3 signaling pathway. *Front Pharmacol* **12**, 615506, doi:10.3389/fphar.2021.615506 (2021).
- 45 Jang, Y. J., Kim, W. K., Han, D. H., Lee, K. & Ko, G. Lactobacillus fermentum

- species ameliorate dextran sulfate sodium-induced colitis by regulating the immune response and altering gut microbiota. *Gut Microbes* **10**, 696-711, doi:10.1080/19490976.2019.1589281 (2019).
- 46 Jin, S. *et al.* Low-dose penicillin exposure in early life decreases Th17 and the susceptibility to DSS colitis in mice through gut microbiota modification. *Sci Rep* **7**, 43662, doi:10.1038/srep43662 (2017).
- 47 Peng, H. *et al.* Tristetraprolin regulates TH17 cell function and ameliorates DSS-induced colitis in mice. *Front Immunol* **11**, 1952, doi:10.3389/fimmu.2020.01952 (2020).
- 48 Chang, Y. Y. *et al.* P2Y1R ligation suppresses Th17 cell differentiation and alleviates colonic inflammation in an AMPK-dependent manner. *Front Immunol* **13**, 820524, doi:10.3389/fimmu.2022.820524 (2022).
- 49 Zhou, J. *et al.* BLT1 in dendritic cells promotes Th1/Th17 differentiation and its deficiency ameliorates TNBS-induced colitis. *Cell Mol Immunol* **15**, 1047-1056, doi:10.1038/s41423-018-0030-2 (2018).
- 50 Liu, Y., Beyer, A. & Aebersold, R. On the dependency of cellular protein levels on mRNA abundance. *Cell* **165**, 535-550, doi:10.1016/j.cell.2016.03.014 (2016).
- 51 Wang, X. *et al.* N⁶-methyladenosine modulates messenger RNA translation efficiency. *Cell* **161**, 1388-1399, doi:10.1016/j.cell.2015.05.014 (2015).
- 52 Lin, S., Choe, J., Du, P., Triboulet, R. & Gregory, R. I. The m⁶A methyltransferase METTL3 promotes translation in human cancer cells. *Mol Cell* **62**, 335-345, doi:10.1016/j.molcel.2016.03.021 (2016).
- 53 Meyer, K. D. *et al.* 5' UTR m⁶A promotes cap-independent translation. *Cell* **163**, 999-1010, doi:10.1016/j.cell.2015.10.012 (2015).
- 54 Tang, Y. *et al.* OsNSUN2-mediated 5-methylcytosine mRNA modification enhances rice adaptation to high temperature. *Dev Cell* **53**, 272-286 e277, doi:10.1016/j.devcel.2020.03.009 (2020).
- 55 Echevarria-Vargas, I. M. *et al.* Co-targeting BET and MEK as salvage therapy for MAPK and checkpoint inhibitor-resistant melanoma. *EMBO Mol Med* **10**, doi:10.15252/emmm.201708446 (2018).
- 56 Kaur, A. *et al.* Remodeling of the collagen matrix in aging skin promotes melanoma metastasis and affects immune cell motility. *Cancer Discov* **9**, 64-81, doi:10.1158/2159-8290.CD-18-0193 (2019).
- 57 Ouyang, S. *et al.* Akt-1 and Akt-2 differentially regulate the development of experimental autoimmune encephalomyelitis by controlling proliferation of thymus-derived regulatory T cells. *J Immunol* **202**, 1441-1452, doi:10.4049/jimmunol.1701204 (2019).
- 58 Pandey, S. P. *et al.* Pegylated bisacycloxypropylcysteine, a diacylated lipopeptide ligand of TLR6, plays a host-protective role against experimental *Leishmania major* infection. *J Immunol* **193**, 3632-3643, doi:10.4049/jimmunol.1400672 (2014).

REVIEWER COMMENTS

Reviewer #1 (Remarks to the Author):

The authors responded very well to my comments. No further comments/remarks.

Reviewer #2 (Remarks to the Author):

I appreciate the authors' hard work in addressing my previous concerns, which addressed some of my concerns but also raised a few more concerns as listed below:

1) Fig 4J and Extended Fig 13 h, the percentages of IL-17A+ CD4 T cells and IFN γ + CD4 cells in the colon are less than 0.5%, which is too low to be true. In the CD45^{hi} CD4 T cell transfer model, the levels of Th17 and Th1 cells in colonic LP are quite high, which is why they mediate intestinal inflammation. Please double-check your experimental procedure/reagents/FACS machine setting/gating strategy for analysis for those experiments. Please show FACS profiles. Only bar charts are not acceptable.

2) Extended Fig 14. The experimental design was not appropriate. The crucial untreated controls were missing. You actually need 5 groups for this purpose: 1-WT to Rag mice with antibody carrier; 2-WT to Rag mice with anti-IL-17A; 3-WT to Rag mice with anti-IL-17F; 4-WT to Rag mice with anti-IL-17A and anti-IL-17F; 5-KO to Rag mice.

3) Please state how many times the experiments have been done in all Figures.

Reviewer #3 (Remarks to the Author):

Authors reply satisfactorily to all my comments

Point-by-point responses to the reviewers' comments

1. Summary of major comments from the reviewers.

We have followed the insightful suggestions and comments from the reviewers to design and perform further experiments and analyses. We are very grateful to the editors and reviewers for their supporting and helpful suggestions which have further improved the quality and potential impact of our work. The major revised results are summarized and listed in the following **Rebuttal Table 1**.

Rebuttal Table 1. Summary of revision results for major comments from the reviewers.

	Question	Clarification on the original submission data	Performed new analyses	Performed new experiments
1	The percentages of IL-17A ⁺ CD4 T cells and IFN γ ⁺ CD4 cells in the colon are less than 0.5%, which is too low to be true.	N/A	In the original Fig. 4j and original Extended Fig. 13h , we analyzed the frequency of IFN- γ ⁺ CD4 ⁺ T, IL-4 ⁺ CD4 ⁺ T, IL-17A ⁺ CD4 ⁺ T and Foxp3 ⁺ CD4 ⁺ T cells against total colonic lamina propria cells (LPs) including non-immune cells, so the percentages showed less than 0.5%. We have now re-analyzed the frequency of T cell subsets against CD4 ⁺ T cell of colonic LPs, and found that the percentage of IL-17A ⁺ CD4 ⁺ T cells is about 5 % and IFN- γ ⁺ CD4 ⁺ T cells is about 17 % in wild-type colitis group (Rebuttal Fig. 1; Revised Fig. 4j and Revised Extended Data Fig. 13h), which is within the range of the previous published T-cell transfer colitis models showing about 3-10 % for	N/A

			IL-17A ⁺ CD4 ⁺ (Th17 cells) and 8-35 % for IFN- γ ⁺ CD4 ⁺ (Th1 cells) in CD4 ⁺ T cells of colonic LPs ¹⁻⁶ .	
2	Extended Fig 14. The experimental design was not appropriate. The crucial untreated controls were missing. You actually need 5 groups for this purpose: 1- WT to Rag mice with antibody carrier; 2- WT to Rag mice with anti-IL-17A; 3-WT to Rag mice with anti-IL-17F; 4- WT to Rag mice with anti-IL-17A and anti-IL-17F; 5-KO to Rag mice.	The experiments of original Extended Fig. 13 and original Extended Fig. 14 were conducted on the same groups of Rag1 ^{-/-} mice with different treatments, so the controls were only included in Extended Fig. 13 . In this revised version, we have included this information in the Figure legend of Revised Extended Fig. 14a-c .	N/A	Due to the severe impact on our research by the COVID pandemic, it becomes infeasible to collect enough numbers of Rag1 ^{-/-} mice to address this comment. Fortunately, our results in the previous submission have shown consistent phenotypes in colitis development upon Nsun2 deletion between DSS-induced colitis and CD4 ⁺ CD25 ⁻ CD45Rb ^{hi} T cell transfer mouse models. As such, we instead employed DSS colitis model to investigate the function of IL-17A/IL-17F. We have followed the reviewer's suggestion to include five experimental groups of wild-type + PBS, Nsun2 ^{ckO} + PBS, wild-type + anti-IL-17A, wild-type + anti-IL-17F, and wild-type + anti-IL-17A/anti-IL-17F to examine the alleviation of colitis development in wild-type mice by blockage of IL-17A or IL-17F. Further supporting evidence was also acquired by testing the colitis development in DSS-induced colitis model of Nsun2 ^{ckO} mice by injecting recombinant mouse-IL-17A (rIL-17A) or IL-17F (rIL-17F), including five experimental groups: wild-type + PBS, Nsun2 ^{ckO} + PBS, Nsun2 ^{ckO} + rIL-17A, Nsun2 ^{ckO} + rIL-17F, and Nsun2 ^{ckO} + rIL-17A/rIL-17F. The results showed that neutralization of IL-17A, IL-17F, or both substantially alleviated the DSS-induced colitis development in wild type mice, with more obvious effect when simultaneously blocking both (Rebuttal Fig. 2a-f, revised Extended Data Fig. 15a-f). In contrast, the alleviation of DSS-induced colitis progression in Nsun2 ^{ckO} mice could be reversed by injecting rIL-17A, rIL-17F, or both (Rebuttal Fig. 3a-f, revised Extended Data Fig. 16a-f). Collectively, these findings demonstrate that both IL-17A and IL-17F serve as the downstream targets of Nsun2 in Th17 cells and participate in the progression of colitis.

REVIEWER COMMENTS

Reviewer #1 (Remarks to the Author):

The authors responded very well to my comments. No further comments/remarks.

Response: We highly appreciate the reviewer for his/her valuable comments and recognition of our work.

Reviewer #2 (Remarks to the Author):

I appreciate the authors' hard work in addressing my previous concerns, which addressed some of my concerns but also raised a few more concerns as listed below:

1) Fig 4J and Extended Fig 13 h, the percentages of IL-17A⁺ CD4 T cells and IFN γ ⁺ CD4 cells in the colon are less than 0.5%, which is too low to be true. In the CD45^{hi} CD4 T cell transfer model, the levels of Th17 and Th1 cells in colonic LP are quite high, which is why they mediate intestinal inflammation. Please double-check your experimental procedure/reagents/FACS machine setting/gating strategy for analysis for those experiments. Please show FACS profiles. Only bar charts are not acceptable.

Response: Thanks for this very thoughtful advice. In the **original Fig. 4j** and **original Extended Fig. 13h**, we analyzed the frequency of IFN- γ ⁺ CD4⁺ T, IL-4⁺ CD4⁺ T, IL-17A⁺ CD4⁺ T and Foxp3⁺ CD4⁺ T cells against total colonic lamina propria cells (LPs) including non-immune cells, so the percentages showed less than 0.5%. We have now re-analyzed the frequency of T cell subsets against CD4⁺ T cell of colonic LPs, and found that the percentage of IL-17A⁺ CD4⁺ T cells is about 5 % and IFN- γ ⁺ CD4⁺ T cells is about 17 % in wild-type colitis group (**Rebuttal Fig. 1; Revised Fig. 4j and Revised Extended Data Fig. 13h**), which is within the range of the previous published T-cell transfer colitis models showing about 3-10 % for IL-17A⁺ CD4⁺ (Th17 cells) and 8-35 % for IFN- γ ⁺ CD4⁺ (Th1 cells) in CD4⁺ T cells of colonic LPs¹⁻⁶.

Rebuttal Fig. 1 a, Gating strategies for flow cytometry analysis of T cell subsets in colonic lamina propria lymphocytes in CD4⁺CD25⁻CD45Rb^{hi} adoptive transfer mouse colitis. **b, c**, The flow cytometry (**b**) and statistical (**c**) analysis of IFN γ ⁺ CD4⁺ T, IL-4⁺ CD4⁺ T, IL-17A⁺ CD4⁺ T and Foxp3⁺ CD4⁺ T cells frequency in LP CD4⁺ T cells from recipient mice transferring CD4⁺CD25⁻CD45Rb^{hi} naïve T cells of wild-type or *Nsun2*^{CKO}. Five animals in each group were analyzed and the experiment was performed twice independently, and representative images are shown. Data are the mean \pm s.e.m ($n=3$ independent experiments). The *P* values were determined using a two-sided unpaired Student's *t*-test (**c**).

Rebuttal Table 2. The percentage of IL-17A⁺ CD4⁺ and IFN γ ⁺ CD4⁺ T cells in lamina propria CD4⁺ T cells.

	Colitis model	Percentage of IL-17A⁺ CD4⁺ T cell (Th17 cells) in LP CD4⁺ T cells	Percentage of IFN-γ⁺ CD4⁺ T cell (Th1 cells) in LP CD4⁺ T cells	Reference
Pedersen's lab	CD4 ⁺ CD25 ⁻ T cell-transplanted mice	4.7-6.8 %	19.7-24.8 %	Schmidt, et al. Inflamm Bowel Dis , 2013 ¹ .
Dietrich's lab	CD4 ⁺ CD25 ⁻ CD45Rb ^{hi} T cell - transplanted in Rag2 ^{-/-} mice	8 \pm 2 %	25 \pm 2 %	Basso, et al. J Gastroenterol , 2018 ² .
Croituru's lab	CD4 ⁺ CD45Rb ^{hi} T cell-transplanted in severe combined immunodeficient mice.	~ 10 %	~ 8 %	Forster, et al. Gastroenterology . 2012 ³ .
Kelsall's lab	CD4 ⁺ CD45Rb ^{hi} T cell-transplanted in Rag2 ^{-/-} mice	3-5 %	20-25 %	Valatas, et al. Mucosal Immunol , 2013 ⁴ .
Sugimoto's lab	CD4 ⁺ CD25 ⁻ CD45Rb ^{hi} T cell-transplanted in severe combined immunodeficient mice.	5.48 \pm 0.74 %	33.64 \pm 2.62 %	Tani, et al. Inflamm Bowel Dis , 2017 ⁵ .
Okada's lab	CD4 ⁺ CD45Rb ^{hi} T cells in severe combined immunodeficient mice.	~ 6 %	~ 22 %	Takahara, M. et al. Sci Rep , 2019 ⁶ .

2) Extended Fig 14. The experimental design was not appropriate. The crucial untreated controls were missing. You actually need 5 groups for this purpose: 1-WT to Rag mice with antibody carrier; 2-WT to Rag mice with anti-IL-17A; 3-WT to Rag mice with anti-IL-17F; 4-WT to Rag mice with anti-IL-17A and anti-IL-17F; 5-KO to Rag mice.

Response: Thanks for the insightful comments and constructive suggestion from the reviewers.

We are sorry for not clearly presenting this data. Actually, the experiments of **original**

Extended Fig. 13 and **original Extended Fig. 14** were conducted on the same groups of *Rag1*^{-/-} mice with different treatments, so the controls were only included in **Extended Fig. 13 d, e, and g**. In this revised version, we have included this information in the Figure legend of **Revised Extended Fig. 14**.

For the group experiment design in the 1st revision, we mainly followed the reviewer's comment about the inconsistent function of the lack of IL-17A in the development of colitis, and tested the function of IL-17A blockage in T-cell transfer colitis model.

In this revised version, the reviewer suggested to test the function of IL-17A and IL-17F using CD4⁺CD25⁻CD45Rb^{hi} T cell transfer colitis model. However, due to the severe impact on our research by the COVID pandemic, it becomes infeasible to collect enough numbers of *Rag1*^{-/-} mice to address this comment. Fortunately, our results in the previous submission have shown consistent phenotypes in colitis development upon *Nsun2* deletion between DSS-induced colitis and CD4⁺CD25⁻CD45Rb^{hi} T cell transfer mouse models. As such, we instead employed DSS colitis model to investigate the function of IL-17A/IL-17F. We have followed the reviewer's suggestion to include five experimental groups of wild-type + PBS, *Nsun2*^{ckO} + PBS, wild-type + anti-IL-17A, wild-type + anti-IL-17F, and wild-type + anti-IL-17A/anti-IL-17F to examine the alleviation of colitis development in wild-type mice by blockage of IL-17A or IL-17F. Further supporting evidence was also acquired by testing the colitis development in DSS-induced colitis model of *Nsun2*^{ckO} mice by injecting recombinant mouse-IL-17A (rIL-17A) or IL-17F (rIL-17F), including five experimental groups: wild-type + PBS, *Nsun2*^{ckO} + PBS, *Nsun2*^{ckO} + rIL-17A, *Nsun2*^{ckO} + rIL-17F, and *Nsun2*^{ckO} + rIL-17A/rIL-17F. The results showed that neutralization of IL-17A, IL-17F, or both substantially alleviated the DSS-induced colitis development in wild type mice, with more obvious effect when simultaneously blocking both (**Rebuttal Fig. 2a-f, revised Extended Data Fig. 15a-f**). In contrast, the alleviation of DSS-induced colitis progression in *Nsun2*^{ckO} mice could be reversed by injecting rIL-17A, rIL-17F, or both (**Rebuttal Fig. 3a-f, revised Extended Data Fig. 16a-f**). Collectively, these findings demonstrate that both IL-17A and IL-17F serve as the downstream targets of *Nsun2* in Th17 cells and participate in the progression of colitis.

Previous findings showed that just like IL-17A, the function of IL-17F in colitis development is also debatable. Dong et al. reported that IL-17F deficiency reduced the DSS-induced colitis⁷. Also, Iwakura et al. demonstrated that blockage of IL-17F suppressed the development of colitis in both DSS-induced colitis and T cell-transferring colitis models^{8,9}. Moreover, Pedersen et al. showed that simultaneously neutralizing both IL-17A and IL-17F *in vivo* suppressed development of CD4⁺CD25⁻ T-cell transfer colitis, whereas neutralization of IL-17A or IL-17F alone was inefficient¹. Consistently, Neurath et al. demonstrated that blockage both IL-17A and IL-17F relieves the development colitis, and proposed that combination of anti-IL-17A and anti-IL-17F might be developed as therapeutics for chronic colitis¹⁰. In Th17-cell related inflammatory diseases, including psoriasis and psoriatic arthritis (PsA), dual blockade of IL-17A and IL-17F has been shown to be more effective than IL-17A blockade alone¹¹. In support, our findings demonstrated that blockage of both IL-17A and IL-17F had the most prominent effect on alleviation of colitis development, even though blockage each factor alone also showed beneficial outcome.

As for the experimental mouse model, we sincerely hope and appreciate the understanding and supporting from the reviewer.

Rebuttal Fig. 2 | Neutralization of IL-17A and IL-17F alleviates the colitis development in DSS-induced mice.

a-f, The wild-type and *Nsun2^{CKO}* mice were treated with 3% DSS in drinking water after blockage of IL-17A (250 μ g/mouse) or IL-17F (250 μ g/mouse) alone, or both signaling with antibodies. Five mice were used in each group. Body weight changes (**a**), representative image of spleen (**b**), macrograph of colons (**c**), the colon length (**d**), hematoxylin-eosin (H&E) staining indicated the severity of colons damage and inflammatory infiltration (**e**), and colitis score (**f**) ($n=5$ independent experiments). Data are the mean \pm s.e.m ($n=5$ independent experiments). The P values were determined by using Two-Way ANOVA test (**a**) and two-sided unpaired Student's t -test (**d, f**). Error bars represent mean \pm s.e.m.

Rebuttal Fig. 3 | Replenishment of rIL-17A and rIL-17F in *Nsun2*-deletion mice promotes colitis development in DSS-induced mice.

a-f, The wild-type and *Nsun2^{ckO}* mice were treated with 3% DSS in drinking water. *Nsun2^{ckO}* mice were injected with recombinant mouse-IL-17A (rIL-17A) (500 ng/mouse) or IL-17F (rIL-17F) (500 ng/mouse) alone, or both once every other day. Four mice were used in each group. Body weight changes (**a**), representative image of spleen (**b**), macrograph of colons (**c**), the colon length (**d**), hematoxylin-eosin (H&E) staining indicated the severity of colons damage and inflammatory infiltration (**e**), and colitis score (**f**) ($n = 5$ independent experiments). Data are the mean \pm s.e.m ($n=4$ independent experiments). The P values were determined by using Two-Way ANOVA test (**a**) and two-sided unpaired Student's t -test (**d, f**). Error bars represent mean \pm s.e.m.

3) Please state how many times the experiments have been done in all Figures.

Response: Thanks for pointing out this. We have included the times of the experiments in all related Figures' legends (Revised Manuscript: Page 22 lines 866-867, 870, 872-873, 876, 877 and 880-881; Page 23 lines 887, 889, 905 and 907; Page 24 lines 920, 922 and 928; Revised Supplementary Information: Page 2 lines 17; Page 5 lines 37, 40, 45 and 46; Page 7 lines 65, and 67-69; Page 11 lines 125-126; Page 12 lines 132-133).

Reviewer #3 (Remarks to the Author):

Authors reply satisfactorily to all my comments

Response: We are very grateful to the reviewer for his/her thoughtful comments for improving the manuscript and approving our work.

References

- 1 Wedebye Schmidt, E. G. *et al.* TH17 cell induction and effects of IL-17A and IL-17F blockade in experimental colitis. *Inflamm Bowel Dis* **19**, 1567-1576, doi:10.1097/MIB.0b013e318286fa1c (2013).
- 2 Basso, L. *et al.* T-lymphocyte-derived enkephalins reduce Th1/Th17 colitis and associated pain in mice. *J Gastroenterol* **53**, 215-226, doi:10.1007/s00535-017-1341-2 (2018).
- 3 Forster, K. *et al.* An oral CD3-specific antibody suppresses T-cell-induced colitis and alters cytokine responses to T-cell activation in mice. *Gastroenterology* **143**, 1298-1307, doi:10.1053/j.gastro.2012.07.019 (2012).
- 4 Valatas, V. *et al.* Host-dependent control of early regulatory and effector T-cell differentiation underlies the genetic susceptibility of RAG2-deficient mouse strains to transfer colitis. *Mucosal Immunol* **6**, 601-611, doi:10.1038/mi.2012.102 (2013).
- 5 Tani, S. *et al.* Digoxin attenuates murine experimental colitis by downregulating Th17-related Cytokines. *Inflamm Bowel Dis* **23**, 728-738, doi:10.1097/MIB.0000000000001096 (2017).
- 6 Takahara, M. *et al.* Berberine improved experimental chronic colitis by regulating interferon-gamma- and IL-17A-producing lamina propria CD4(+) T cells through AMPK activation. *Sci Rep* **9**, 11934, doi:10.1038/s41598-019-48331-w (2019).
- 7 Yang, X. O. *et al.* Regulation of inflammatory responses by IL-17F. *J Exp Med* **205**, 1063-1075, doi:10.1084/jem.20071978 (2008).
- 8 Tang, C. *et al.* Suppression of IL-17F, but not of IL-17A, provides protection against colitis by inducing Treg cells through modification of the intestinal microbiota. *Nat Immunol* **19**, 755-765, doi:10.1038/s41590-018-0134-y (2018).
- 9 Hall, A. O., Towne, J. E. & Plevy, S. E. Get the IL-17F outta here! *Nat Immunol* **19**, 648-650, doi:10.1038/s41590-018-0141-z (2018).
- 10 Leppkes, M. *et al.* ROR γ -expressing Th17 cells induce murine chronic intestinal inflammation via redundant effects of IL-17A and IL-17F. *Gastroenterology* **136**, 257-267, doi:10.1053/j.gastro.2008.10.018 (2009).
- 11 Ali, Z. *et al.* Bimekizumab: a dual IL-17A and IL-17F inhibitor for the treatment of psoriasis and psoriatic arthritis. *Expert Rev Clin Immunol* **17**, 1073-1081, doi:10.1080/1744666X.2021.1967748 (2021).

REVIEWERS' COMMENTS

Reviewer #2 (Remarks to the Author):

All my previous concerns have been addressed appropriately

2. Point-by-point responses to reviewers' comments

Reviewer #2 (Remarks to the Author):

All my previous concerns have been addressed appropriately.

Response: We thank the reviewer very much for his/her thoughtful scientific advice during the whole review process.